# Global assessment of sub-national drought impact based on the Geocoded Disasters dataset and land reanalysis

Yuya Kageyama[1], Yohei Sawada[1]

[1]Institute of Engineering Innovation, The University of Tokyo, Tokyo, 113-8656, Japan

*Correspondence to*: Yuya Kageyama (ykageyama21@gmail.com)

**Abstract.** Despite the importance of a linkage between hydro-meteorological drought hazards and their socio-economic impact, the linkage at a sub-national level has yet to be evaluated due to the lack of precise sub-national information on disaster locations. Using the newly developed Geocoded Disasters (GDIS) dataset, we examined whether the sub-national socio-economic drought impact information in GDIS could be represented by hydro-meteorological hazards quantified from
soil moisture in ERA5-Land during 1964–2018. We found that the socio-economic drought impacts shown in GDIS were generally represented by drought hazards quantified from ERA5-Land soil moisture. Our comparison between GDIS and ERA5-Land could benefit the quantification of vulnerability to drought, and we found that Sub-Saharan Africa and South Asia were vulnerable to drought while North America and Europe were less vulnerable to drought. Both GDIS and ERA5-Land indicated that the Horn of Africa, northern China, and western India were drought-prone areas. Since it is difficult for
national-level analyses to accurately identify the locations of drought-prone areas especially in large countries such as China and India, our analysis clarifies the importance of the use of sub-national disaster information.

## 1 Introduction

Drought is one of the costliest natural disasters with cascading impacts on multiple socio-economic sectors (Mishra and Singh, 2010). Wilhite and Glantz (1985) proposed a conceptual model of drought propagation, from natural hydro-
meteorological hazards defined by physical characteristics (e.g., precipitation, soil moisture, or streamflow) to socio-economic drought impacts (e.g., crop yield loss, water shortage, or health problem). The propagation from the natural hydro-meteorological hazards to the socio-economic impact can be affected by many regional vulnerability factors, such as infrastructure, economic, social, or cultural assets (e.g., Fuchs et al., 2019; Lavell et al., 2012; UNDP, 2004; Wilhite and Glantz, 1985). To understand this drought propagation, a sub-national level disaster analysis is necessary, rather than
aggregated national level disaster analyses (Rosvold and Buhaug, 2021). How historical drought events evolved from natural hydro-meteorological hazards to socio-economic drought impacts at a sub-national level needs to be analyzed to improve regional drought mitigation measures.

Several studies have analyzed the linkage between natural hydro-meteorological hazards and socio-economic drought

impacts to quantify the regional characteristics of historical drought events. Disaster databases such as the Emergency Events Database (EM-DAT) (Guha-Sapir et al., 2022), the European Drought Impact report Inventory (EDII) (Stahl et al., 2016), US Drought Impact Reporter (US DIR) (Wilhite et al., 2007), as well as newspaper information (de Brito et al., 2020) have been used as reference data of historical socio-economic impacts. Bachmair et al. (2016) used EDII to estimate the thresholds of hydro-meteorological drought indices at which socio-economic droughts occur in Germany and UK at a sub-

national level. Noel et al. (2020) compared the U.S. Drought Monitor (USDM) (Svoboda et al. 2002), a weekly map depicting severity and spatial extent of drought, with US DIR at the state level. Although EDII and US DIR contain detailed disaster impact information at a sub-national level and are useful to quantify the linkage between hydro-meteorological hazards and socio-economic impacts, they do not cover the entire globe. EM-DAT is a global database and has been extensively used for the international comparison of disaster risks and vulnerability (e.g., Jägermeyr et al., 2018; Shen and

Hwang, 2019; Tschumi and Zscheischler, 2020). Although some studies used text-based disaster locations (i.e., names of affected provinces, districts, and towns) in EM-DAT to perform the sub-national scale analyses, they simply evaluated the applicability of a drought index for the specific regional events (e.g., Bayissa et al., 2018; Lu et al., 2019) and for global events in a short period of time (2010–2015) (Sánchez et al., 2018). The sub-national impact information of the disaster database has not been fully used to quantify the linkage between hydro-meteorological drought hazards and socio-economic

impacts in a global scale. In addition, regional vulnerability against drought events has not been quantified by using such database in a global scale.

Instead of the disaster databases, agricultural production or remotely sensed vegetation dynamics have also been used to assess the impact of drought on society. Udmale et al. (2020) compared cereal production with drought indices such as

Standardized Precipitation Index (SPI) (McKee et al., 1993) and Standardized Precipitation Evaporation Index (SPEI) (Vicente-Serrano et al., 2010) in India. Kim et al. (2019) examined the vulnerability of cereal production to drought in a country scale using a global crop model. Chen et al. (2020) quantified the impact of droughts on vegetation growth for different biome types and climate regimes by comparing SPEI and a vegetation index. Although agricultural production and vegetation dynamics are available globally and easy to be quantified, there are some problems to use them as reference data

of socio-economic drought impacts. Agricultural production can be affected by factors other than drought and it can capture aggregated information on large events (Bachmair et al., 2016). It is unclear whether socio-economic drought impacts are associated with declined vegetation growth. It is ideal to treat the socio-economic drought impact based on a disaster database since it directly shows events in which the society has actually suffered from drought.

The linkage between natural hydro-meteorological hazards and socio-economic drought impact at a sub-national level has yet to be globally evaluated. The major obstacle is a lack of accurate information of socio-economic drought impacts in sub-national scales (Bachmair et al., 2016). Recently, a global dataset of geocoded disaster locations, the Geocoded DISasters

(GDIS), has been developed (Rosvold and Buhaug, 2021). Although EM-DAT contains information about the location of disasters, they are text-based information and some events have incomplete information about their locations, which is not suitable for comprehensive geospatial analyses. GDIS is the geocoded database based on the EM-DAT's location information with some manual validations and provides GIS polygons of affected administrative units. GDIS can be a useful tool to globally assess the linkage between natural hydro-meteorological hazards and socio-economic drought impact with precise locations at a sub-national level.

This study aims to examine the linkage between natural hydro-meteorological hazards and the sub-national socio-economic drought impact shown in GDIS. As a natural hydro-meteorological hazard, we used drought indices generated from soil moisture simulated by land reanalysis, ERA5-Land (Muñoz-Sabater, 2019; Muñoz-Sabater, 2021). First, we examined whether the GDIS drought events were generally represented by the drought indices quantified from ERA5-Land. Then, we quantified the levels of drought indices associated with GDIS drought events in different geographical regions, which could benefit the quantification of vulnerability to drought. Finally, we compared the global spatial distribution of drought-prone areas in GDIS with those quantified from ERA5-Land.

## 2 Data

### 2.1 ERA5-Land

To calculate drought indices, ERA5-Land soil moisture data were used. Wilhite and Glanz (1985) mentioned that soil moisture plays an important role in the drought propagation since it affects both agricultural and hydrological aspects of drought (see also Sawada (2018)). Many drought monitoring systems have also used soil moisture as one of the most important variables (e.g., USDM, Svoboda et al., 2002; The German drought monitor, Zink et al., 2016; InterSucho in Czech Republic and Slovakia, Trnka et al., 2020).

We used monthly averaged data from 1950 to 2020. The original spatial resolution of 0.1° was upscaled to 0.25° to reduce the data volume, by using a remap function of the Climate Data Operators (CDO) version 2.0.0 (Schulzweida, 2021). This spatial resolution is relatively high compared to the previous global-scale drought studies (e.g., Hanel et al., 2018; Herrera-Estrada et al., 2017; Mocko et al., 2021; Sawada, 2018).

We used the first (0–7 cm), second (7–28 cm), and third (28–100 cm) layers' soil moisture in ERA5-Land to generate drought indices. Since previous works used soil moisture from the top to 1–2 m soil depths as root zone soil moisture (e.g., Almendra-Martín et al., 2021; Hanel et al., 2018; Herrera-Estrada et al., 2017; Mocko et al., 2021), we also used the top 1 m (0–100 cm) soil moisture data. For the top 1 m soil moisture, we calculated the weighting average of soil moisture in the first, second, and third layers according to their thicknesses.

## 2.2 GDIS

GDIS (Rosvold and Buhaug, 2021) can be downloaded from https://cmr.earthdata.nasa.gov/search/concepts/C2022273992-SEDAC.html (data downloaded for this study: October 2021). GDIS is generated based on EM-DAT. A natural disaster is recorded into EM-DAT if at least one of the following criteria is fulfilled: 10 or more people dead; 100 or more people affected; the declaration of a state of emergency, and a call for international assistance (Guha-Sapir et al., 2022).

The 282 drought events from 1964 to 2018 were analyzed. Each drought event is distinguished based on the EM-DAT database's event identifier (*disasterno*). In EM-DAT, disaster events are uniquely distinguished by the combination of an 8-digit disaster code and a 3-digit country code. In contrast, GDIS uses only the 8-digit disaster code, which is common with EM-DAT, and assigns the same identifier to a disaster event even if it spreads over multiple countries. In the case of extensive drought events, such as ones induced by El Niño, it is not reasonable to treat distant countries with the same event identifier. In this study, the event classification of the original EM-DAT was adopted, so that events with the same disaster code that spread over multiple countries in GDIS were analyzed as a separate event for each country. Originally, there are 433 drought events in GDIS and 282 events that met the following criteria were used in this study: (1) The drought period is longer than or equal to two months and (2) The GDIS event area is larger than or equal to 50 grid cells in the upscaled ERA5-Land. We did not analyze flash drought, which occurred in shorter than two months. The effective resolution of the phenomena that can be represented by a numerical simulation model is several times as large as the original size of computational grids (Skamarock, 2004), so that the events with the small extent relative to the grid spacing were neglected in this study. GDIS itself does not have drought period information, namely when the event starts and ends. The drought period information was added to GDIS via EM-DAT database's event identifiers. EM-DAT shows only the event year and provides no information on the start and/or end month for some drought events. In such cases, we applied January for the start and December for the end of the event. GDIS provides affected spatial geometry in the form of GIS polygons of administrative units. Administrative units with the same event identifier (*disasterno*) were treated as one "GDIS event area" (see Fig. 2 as an example). Sánchez et al. (2018) treated one drought event per one administrative unit. However, this event classification depends on the fineness of the division of administrative units (e.g., Thailand, where administrative units are very finely divided, has more than 50 events during 2010–2015 in Sánchez et al. (2018)), which affects the results of drought detection skill. Therefore, we treated administrative units with the same event identifier as one drought event, following EM-DAT classification.

## 2.3 Other supporting data

To show the levels of drought indices associated with GDIS drought events by geographical regions, we used the classification of the world bank geographical regions.

As a proxy of exposure data, we used the MODIS land cover climate modeling grid (MCD12C1) version 6 data product (Friedl and Sulla-Menashe, 2015). This land cover product has 17 classes. The temporal resolution is yearly, and we used the latest, 2020 data. The original spatial resolution is 0.05° and we resampled it to 0.25° by the nearest neighbour approach.

## 3 Methodology

### 3.1 Drought indices

We used two drought indices, the Drought Area Percentage (DAP) and the Standardized Deficit Index (SDI), to evaluate the severity of the hydro-meteorological drought hazard in ERA5-Land. For the soil moisture data in each grid cell, percentiles were first calculated for each calendar month separately during 1950–2020. After the percentiles were calculated, only data during the period with the GDIS drought events (1964–2018) was used in all subsequent steps of this study. We used the longer period of original ERA5-Land data (1950–2020) to calculate percentiles than the study period (1964–2018) to yield more robust percentile values. The 20th percentile was taken as a threshold for defining a drought at each grid cell (Sheffield and Wood, 2011; Hanel et al., 2018). DAP is the maximum percentage of the area where soil moisture is below the 20th percentile threshold within the GDIS event area during the GDIS drought period. The higher the percentage means the severer the hydro-meteorological hazard is. DAP has been used as a drought index in many studies (e.g., Sánchez et al. 2018; Udmale et al. 2020).

DAP is a snapshot of the long-lasting drought phenomenon and does not include the cumulative effects of the long-lasting drought. The other limitation of DAP is that it could be affected by the size of the GDIS event area; DAP tends to be small in large event areas. In addition to DAP, we developed a new drought indicator, called SDI, which accounts for the cumulative effects of drought and is less influenced by the size of the GDIS event area. First, a deficit volume, a cumulative deviation below the 20th percentile threshold, was calculated for each grid cell. Then, we summed up the maximum annual deficit volume per grid cell in each GDIS event area, which is defined as the annual maximum deficit volume in the GDIS event area. The cumulative effect of the movement of drought areas can be considered by calculating the annual maximum value for each grid before averaging the values within the GDIS event area. Finally, the annual maximum deficit volume in the GDIS event area was standardized, dividing each year's annual maximum deficit volume by the mean of the annual maximum deficit volume over the period (1964–2018). The higher SDI means the severer hydro-meteorological hazard, and the value of 1 is the standard annual maximum drought event. The standardization makes it possible to compare the different events across space and time, even if the size of the GDIS event area is quite different. Hanel et al. (2018) calculated SDI for each grid cell. We extended this methodology to evaluate the drought index representative in the GDIS event area.

## 3.2 Evaluation of the drought indices by GDIS

To evaluate whether the GDIS drought events are generally represented by the drought indices quantified from ERA5-Land, we tested whether the drought indices during the GDIS drought period were distinguishable from those during the whole period (1964–2018). We applied a bootstrap random resampling method to show the distributions of drought indices for the whole period. For DAP, we set a 12-month moving window, which is approximately the mean of the drought duration in GDIS, and extracted the maximum percentage in each window for each GDIS event. From this assemblies, we extracted DAP randomly with 1000 replications. For SDI, we extracted SDI randomly with 1000 replications from the whole study period. We used two-sample Kolmogorov–Smirnov ($K$–$S$) test (Massey, 1951) to quantify the difference of the distributions of drought indices during the GDIS drought period and the whole period. If the $p$-value of $K$–$S$ test is smaller than 0.01, the hypothesis that two distributions follow the same distribution is rejected at 1% significance level. Due to the difference of the sample size (i.e., Drought period: 282; Whole period by a bootstrap random resampling method: 1000), the distributions were normalized, namely the densities sum to 1, prior to the comparison. We recognized that the GDIS drought events are generally represented by the drought indices quantified from ERA5-Land if the median of the drought index during the GDIS drought periods is higher than that of the whole period and the two distributions of the drought index are not statistically the same.

## 3.3 Regional levels of drought indices associated with GDIS drought events

The levels of hydro-meteorological drought indices associated with drought events shown in GDIS are different in different regions. Vulnerability could explain these differences (Delbiso et al. 2017; Gasparrini et al. 2015; Tschumi and Zscheischler, 2020). Note that vulnerability is not the only explanation for these differences; exposure is another factor that influences the linkage between hazards and impact (Visser et al., 2014; see also the discussion section). Since we did not directly include exposure, we recognized these differences as "the proxy of vulnerability". Following Bachmair et al. (2016), and Tschumi and Zscheischler (2020), the levels of SDI which are associated with drought events in GDIS were quantified and analyzed. The levels of SDI were stratified by geographical regions to understand the distribution of the proxy of vulnerability in each region.

## 3.4 Global drought frequency analysis by drought clustering

We analyzed whether drought-prone areas identified by drought indices are globally consistent with those found in GDIS. We applied the drought clustering method (Andreadis et al., 2005) to search for the spatially contiguous areas (or clusters) under drought at each timestep. In this drought clustering, we assume that drought occurs over a reasonably large spatial area driven by a large-scale climate process (Sheffield and Wood, 2011). We used the processing code developed by Herrera-Estrada and Diffenbaugh (2020).

After the percentiles are calculated in each grid cell, a 2-D median filter is applied to each monthly global data to smooth out small-scale noise. Contiguous areas under drought (soil moisture below the 20th percentile in this study) are aggregated into clusters at each timestep. Following Herrera-Estrada and Diffenbaugh (2020), we analyzed clusters that reach a maximum area of at least 100,000 km$^2$ (approximately 120 grid cells in the upscaled ERA5-Land) to focus on large-scale droughts. For
a sensitivity analysis of this size of drought clusters, see Text S1 and Fig. S1. The location of the cluster centroid is detected at each time step using the weighting average of the cluster's location with the intensity values of the cluster grid cells. Droughts whose centroids fell within the barren or sparsely vegetated areas based on MODIS land cover were masked out from the cluster analysis, due to the little or no exposure (i.e., population, assets) (e.g., Carrão et al., 2016; Herrera-Estrada et al., 2017). We confirmed that there were no drought events in GDIS which were fully included within the barren or sparsely
vegetated areas. Figure 1 demonstrates this drought clustering. The cluster centroid shows the area that experiences higher drought displacement, and we made an upscaled map of cluster centroids from the original spatial resolution of 0.25° to 2.5°. For a sensitivity analysis of this upscale resolution, see Text S2 and Fig. S2. See Andreadis et al. (2005) for details about the clustering method.

We visualized the socio-economic drought-prone areas by overlaying all polygons of the GDIS. We compared the regional drought frequencies in GDIS with the number of drought cluster centroids by ERA5-Land. We examined whether hydro-meteorological drought-prone areas are consistent with those found in GDIS.

## 4 Results

### 4.1 The performance of drought indices to detect the drought

Figure 2 demonstrates DAP and SDI for the drought events in Ethiopia and Argentina in 2009. For both drought indices, higher values indicate severer drought. Figures 2 (a) and 2 (e) show that DAPs in the first and second soil layers respond to rainfall deficit more quickly than the deep soil layers. During the GDIS drought period, all layers show high DAPs, and the third layer has been experiencing a high DAP for a long time. DAPs in the first and second layer are sometimes high outside of the GDIS drought period. Root-zone (0–100 cm) layer generally follows the third layer's fluctuations, though the root-
zone positions between the second and third layers during the GDIS drought period in Ethiopia case. Figures 2 (c) and 2 (g) show that SDIs fluctuate less compared with DAPs. This is because SDI considers the cumulative effect. During the GDIS drought period, all layers show high SDIs, and the differences between the GDIS drought period and the non-drought period stand out more prominently compared with DAP. The third layer shows the highest SDI during the GDIS drought period, especially in Argentina case, reflecting the consistently high DAP during the GDIS drought period. Root-zone (0–100 cm)
layer generally follows the third layer's fluctuations.

Figures 3 and 4 reveal that ERA5-Land based drought indices can distinguish the GDIS drought period from the whole period. The value above the violin plot shows the difference of the median values in the GDIS drought period and the whole period. In Fig. 3, DAP during the GDIS drought periods is significantly higher ($p<0.01$) than that of the whole period in all soil layers. Note that the samples in the whole period shown in Fig. 3 include those during the GDIS drought period. In addition, severe drought events unreported in GDIS may also be included. The difference of the median values of DAP in the GDIS drought period and the whole period is largest in the third layer (28–100 cm) case. In Fig. 4, SDI during the GDIS drought period is significantly higher than that of the whole period in all soil layers, as we found in DAP. Although the second, third and root-zone soil layers show the similar distributions, the difference of the median values of SDI in the GDIS drought period and the whole period is largest in the root-zone (0–100 cm) case. Both of drought indices based on ERA5-Land can generally represent the GDIS drought events. Note that although we confirmed a general linkage between drought hazards and the GDIS drought events, some GDIS events could not explained by our indices based on the anomaly of soil moisture. We will use SDI for the regional comparison shown below, because SDI is a standardized indicator, which allows the comparison between the different events across space and time, even if the size of the GDIS event area is substantially different.

## 4.2 Regional levels of drought indices associated with GDIS drought events

Figure 5 shows the distribution of the root-zone layer's soil moisture-based SDI stratified by geographical regions. The colour of the figure shows the average soil moisture over the study period. Sub-Saharan Africa and South Asia have many small SDI events associated with the GDIS identified drought, while North America and Europe have a large number of large SDI events. Having many small SDI events indicates that less severe hydro-meteorological droughts have caused serious socio-economic impacts, meaning that the regions are vulnerable to drought. On the other hand, the regions with many large SDI events can be recognized as less vulnerable regions to drought. Thus, Sub-Saharan Africa and South Asia are vulnerable to drought, while North America and Europe are less vulnerable to drought. Sub-Saharan Africa, which is vulnerable to drought, shows lower water availability. This regional characteristic of the proxy of vulnerability to drought can be found when SDI is generated by soil moisture in different soil layers (not shown). Note that Middle East & North Africa were excluded from the analysis because the sample size was too small ($n = 4$).

## 4.3 Global drought frequency analysis by drought clustering

Figure 6 shows the number of drought events at a sub-national level during 1964–2018 based on GDIS. It shows that the Horn of Africa, Mozambique, northern China, and western India are socio-economic drought-prone areas. Each region is shown enlarged in Figs. 6 (b) to 6 (e). Figure 7 shows the number of drought events on the aggregated national level during the same period based on EM-DAT. Although we can see that the number of drought events is high in China, there is little information about the regional differences in drought-prone areas.

This distribution of drought-prone areas in GDIS can be reproduced by ERA5-Land. Figure 8 shows the number of the drought cluster centroids upscaled to 2.5° based on drought clusters from ERA5-Land third layer's soil moisture. Drought-prone areas quantified from ERA5-Land soil moisture (Fig. 8) are consistent with those listed in GDIS (Fig. 6). The Horn of Africa, northern China, and western India can also be recognized as drought-prone areas by ERA5-Landbased drought clusters. Mozambique cannot be identified as a drought-prone area in ERA5-Land. Note that the number of the drought

cluster centroids (Fig. 8) would be larger than the number of drought events in GDIS (Fig. 6). The number of drought events in GDIS is counted as one event even if a GDIS event lasts several months. On the other hand, the number of drought cluster centroids is counted in every monthly time step. Several clusters may be contained simultaneously in a large GDIS drought area. ERA5-Land identifies some drought-prone areas which are not included in GDIS, such as Namibia, Indonesia, and Spain. See also the supplement material for sensitivity analysis with different thresholds of the size of drought clusters (Fig.

S1), showing that drought-prone areas found in GDIS cannot be reproduced by ERA5-based drought-prone areas when we used too small or large thresholds of the size of drought clusters. The locations of drought-prone areas are almost the same when drought clusters are generated by soil moisture in different soil layers (Fig. S3). The drought-prone areas are most distinguishable from their surroundings in the third layer case.

## 5 Discussion

In previous studies, the verification of sub-national drought events by hydro-meteorological data has been insufficient. There are some works only on the specific regions (e.g., Bayissa et al., 2018; Lu et al., 2019) or in a short period of time (Sánchez et al., 2018), due to the lack of precise sub-national information on disaster locations. Using the latest sub-national disaster database, GDIS, this study was able to cover a large number of drought events compared to previous studies. In Sánchez et al. (2018), the criterion for the detection of drought events was that more than one-third of the area was under drought. However,

the size of the drought event area could affect the criterion, and the threshold of one-third is rather subjective. By defining the standardized drought index, this study uniformly and objectively showed the representation of sub-national drought information by ERA5-Land soil moisture, even if the size of the event differs.

The comparison of SDI associated with GDIS drought events across regions benefits the quantification of vulnerability to

275 drought in each region. We confirmed that Sub-Saharan Africa and South Asia were vulnerable to drought, while North America and Europe were less vulnerable to drought. Tschumi and Zscheischler (2020) also showed smaller climate anomalies in less developed countries associated with EM-DAT disasters, meaning that less developed countries were vulnerable to natural hazards, as shown in our Fig. 5. Previous studies have shown that higher GDP per capita is associated with lower vulnerability to natural hazards (e.g., Kim et al., 2019; Tanoue et al., 2016). North America and Europe are high-

280 income countries, and these previous works support our findings. There are global vulnerability indices such as the

WorldRiskIndex (Welle and Birkmann, 2015), INFORM index (Marin-Ferrer et al., 2017), and ND-GAIN (Chen et al. 2015), which combine socio-economic factors such as economic level, infrastructure level, and education level. These indices have also indicated that Sub-Saharan Africa and South Asia are vulnerable, while North America, Europe, Australia, and Japan are less vulnerable to natural hazards (Birkmann et al., 2021; Birkmann et al., 2022; Garschagen et al., 2021). The reason why the low-income countries are vulnerable to drought could be the lack of drought mitigation measures (e.g., dams, irrigation system, early-warning system, etc.), as pointed out in previous studies (e.g., Lavell et al., 2012; Stringer et al., 2020; UNEP, 2018). As shown in Fig. 5, Sub-Saharan Africa, which was vulnerable to drought, showed lower water availability. It may be another reason for the difficulty in managing the drought hazards in Sub-Saharan Africa.

GDIS, a sub-national level disaster locations dataset, has enabled us to understand drought-prone areas on a finer scale than the previous global-scale analyses. EM-DAT is generally a national-level database with limited sub-national disaster information. Shen and Hwang (2019) compared the frequency of disaster occurrence in EM-DAT at the national level and pointed out that frequent areas were large or populated countries. GDIS provides more detailed information about drought-prone areas, especially in large countries such as China and India. We successfully clarified that there was the considerable heterogeneity of the drought-prone areas within the country.

There were some inconsistencies between hydro-meteorological drought-prone areas in ERA5-Land and socio-economic drought-prone areas in GDIS. Mozambique is a socio-economic drought-prone area in GDIS, which cannot be identified as a drought-prone area in ERA5-Land. Madagascar, which is geographically closer to Mozambique, is a drought-prone area in ERA5-Land. The performance of ERA5-Land to simulate soil moisture might affect these inconsistencies. In contrast, there were some hydro-meteorological drought-prone areas in ERA5-Land, which were not included in socio-economic drought-prone areas in GDIS (e.g., Spain, Namibia, and Indonesia). Spain, a member of European countries, is less vulnerable to drought, as shown in Fig. 5 (two events were observed in Spain, and their average SDI was 4.3). In Namibia, a lack of exposure makes socio-economic droughts less likely to occur. When assessing socio-economic impact, the presence of the exposure should also be considered (Visser et al., 2014). Namibia has extremely low population density throughout the country (under 1 person per km$^2$ in 2020, Gridded Population of the World (GPW) version 4.11, Doxsey-Whitfield et al., 2015). Similarly, western Australia, central and eastern Russia do not have socio-economic droughts in GDIS due to the low population density. In Indonesia, the absolute amount of rainfall is so high that the relatively small soil moisture may not cause socio-economic drought. Kim et al. (2019) reported that there was no clear correlation between drought severity and yields reduction in areas where average annual precipitation is more than 900 mm. Indonesia is one of the rainiest regions on the globe, with more than 2,700 mm annual precipitation (FAO, 2022). Despite these individual circumstances, our results showed that socio-economic drought-prone areas in GDIS were generally consistent with hydro-meteorological drought-prone areas in ERA5-Land (the Horn of Africa, northern China, and western India), indicating that the reanalysis product can be utilized to show a "potential" of socio-economic drought impact.

The consistency between hydro-meteorological drought-prone areas in ERA5-Land and socio-economic drought-prone areas in GDIS shows that spatially large hydro-meteorological droughts (we analyzed at least 100,000 km$^2$) typically lead to impacts as shown in GDIS. Although the drought frequency defined by simulated soil moisture is the same everywhere at the grid level (we set the 20th percentile as a drought threshold), there was considerable heterogeneity in the spatially large

drought-prone areas (Fig. 8). There are some factors that contribute to the emergence of drought-prone areas, such as El Niño Southern Oscillation (ENSO), La Niña, intertropical convergence zone (ITCZ), monsoon, landatmosphere coupling, and anticyclones (Christian et al., 2021). La Niña affects the Horn of Africa, northern China, and western India and has caused severe drought impacts (Funk, 2011; Jain et al., 2021). Ummenhofer et al. (2011) clarified the effect of El Niño–Indian monsoon relationship on drought in western India. Spatio-temporally large events such as La Niña might cause

drought to persist, which leads to drought impacts as shown in GDIS. However, the drought factors are complex, and much future work is needed to reveal the mechanism of the emergence of drought-prone areas.

Although various reanalysis products have been developed and their validations have been conducted by comparing them with earth observation data (e.g., Muñoz-Sabater et al., 2021; Reichle et al., 2017; Rodell et al., 2004), few studies have

330 examined the validation in terms of the disaster occurrence. Sawada (2018) compared the areas identified as drought from a reanalysis product with the disaster records from EM-DAT, but only in a country-scale. As seen in Fig. 7, national-level information does not provide accurate views of disaster locations, so that it is insufficient for validation data. The use of sub-national disaster databases such as GDIS opens the door to validate reanalysis products in terms of the disaster occurrence.

Although there are many variables to quantify hydro-meteorological droughts, we showed that soil moisture could represent the GDIS drought events in time and space. In the comparison of the soil layers, deep layers (i.e., the third layer (28–100 cm) and root-zone layer (0–100 cm)) were affected for a longer period, which made SDI tend to be higher than that of the first (0–7 cm) and second (7–28 cm) layers during drought. In drought clustering, the drought-prone areas were most distinguishable from their surroundings in the third layer case. Sawada and Koike (2016) used land reanalysis products to

confirm that drought propagates from surface to root-zone (5–100 cm) soil moisture and then to vegetation, and showed that root-zone soil moisture and vegetation were good indices to represent the prolonged drought impact in the case of the Horn of Africa drought (2010–2011). In this study, we confirmed that many of the serious drought events such as those listed in GDIS were the events that were associated with the soil moisture deficit not only on the surface layer but also down to the root. Many drought studies have used root-zone soil moisture and our study has reinforced its validity. Hao and Singh (2015)

suggested that a single drought index is insufficient to capture different impact types of droughts (e.g., water shortage, famine. wildfire, etc.). Several studies have tried to develop a new combined drought index based on several hydro-meteorological variables (e.g., precipitation and soil moisture) to express socio-economic drought impact by using Random Forest models (e.g., Bachmair et al., 2016; Hobeichi et al., 2022). We used the percentile soil moisture, deviation from the

normal condition, to quantify drought following many previous studies (e.g., Sheffield and Wood, 2011; Hanel et al., 2018).

However, the inconsistency for Indonesia between hydro-meteorological drought-prone areas and drought-prone areas found in GDIS implies that the drought in extremely wet regions might not be well represented. It means that our drought quantification method based on relative values of soil moisture cannot accurately consider the amount of regularly available water resources. An alternative way to quantify drought is to use an absolute soil moisture value, but it is not straightforward to quantify drought events by absolute soil moisture values. The thresholds of drought impact occurrences in absolute values

are different in different regions, because ecosystems/societies have adapted to the water availability in their region. It means that a unified drought analysis across multiple regions is difficult to be developed based on absolute values of soil moisture. The other limitation is the biases of absolute values of soil moisture in reanalysis products. Many climate studies have used relative values rather than absolute values because biases in climate models are less important in relative values (Liu and Key, 2016). Further studies are needed for the variable selection and drought indices according to the type of drought, which

leads to more accurate representation of socio-economic drought impact by hydro-meteorological variables.

The relationship between hazards and impact is much more complex than addressed in this study. Many studies have revealed the non-linear relationships between the drought severity and the reduction of vegetation growth (e.g., Chen et al. 2020; Meyer et al., 2014), where damage increases suddenly when the drought severity exceeds a certain critical threshold.

On the other hand, de Brito et al. (2020) reported that there was a linear relationship between the drought severity and the number of drought articles as a proxy of socio-economic drought impacts. In addition, drought is a long-lasting disaster and there is a time-lag between hazards and impact, so that the period of hydro-meteorological drought is not necessarily consistent with the period considered as a disaster in EM-DAT. Some studies have revealed that the impacts of drought last even after the hydro-meteorological drought ends (e.g., Shahbazbegian and Bagheri, 2010). Yokomatsu et al. (2020)

analyzed the impact of the drought in terms of the economic development after the drought. In any case, further analyses are needed to focus on the chronological correspondence to drought hazards.

The limitation of the quantification of the proxy of vulnerability in this study is that we only captured the static conditions over time. We do not reveal which factors (e.g., infrastructure, economic, social, or cultural assets) contribute to the

375 vulnerability. Vulnerability to disasters is complex and dynamic. For example, people's water demand could dynamically change after experiencing drought events (Gonzales and Ajami, 2017). Improved irrigation scheduling (Cao et al., 2019) and dam operation (Wu et al., 2018) based on forecasts could reduce the drought damage. Exposure is another important factor that influences the linkage between hazards and impact (Visser et al., 2014). The inconsistency between hydro-meteorological drought-prone areas in ERA5-Land and drought-prone areas found in GDIS in Namibia implies that the

380 quantification of exposure is necessary to strengthen the analysis in our study. It is necessary to identify what has been damaged (e.g., people, crops, forests, etc.) to quantify exposure. However, EM-DAT provides no information about impact types (e.g., water shortage, famine, wildfire, etc.) in many drought events, which inhibits the identification of what has been

damaged. Like vulnerability, exposure is complex and dynamic. For example, the level of exposure is affected by changes in crop growing with the seasons (Bodner et al., 2015). To improve our analysis on vulnerability shown in Sect. 4.2, detailed analyses on the complex and dynamic nature of both vulnerability and exposure are necessary.

The major limitation of this study is the incompleteness of the drought impact data. Although GDIS enables sub-national drought analysis, GDIS only covers about 60% of droughts in EM-DAT due to vague or unknown location names in EM-DAT (Rosvold and Buhaug, 2021). Note that even EM-DAT does not cover all disasters. Moreover, EM-DAT and GDIS have insufficient quantitative impact information. EM-DAT provides no information about the amount of damage in multiple drought events. There is a lot of uncertainty in the amount of damage because it is difficult to quantify indirect damages of drought (e.g., Yokomatsu et al., 2020). Although we excluded GDIS drought events shorter than two months from our analysis, some of the analyzed events might be shorter than two months. This is because we applied January for the start and December for the end of the event if the start and/or end months of events shown in EM-DAT were unclear. Although GDIS is a pioneering work to achieve the detailed analysis of the relationship between hydro-meteorological drought hazards and socio-economic impact of drought in a global scale, which we performed in this paper, there is much room for improvement of the global disaster database such as including detailed and quantifiable damage information by following the approaches of EDII and US DIR.

**6 Conclusions**

We evaluated how the sub-national socio-economic drought impact information shown in GDIS could be reproduced by the natural hydrological drought indices generated by the reanalysis product, ERA5-Land. We confirmed that the reanalysis product represented the socio-economic drought impacts in GDIS at a statistically significant level. Our comparison between GDIS and ERA5-Land could benefit the quantification of vulnerability to drought, and we showed that Sub-Saharan Africa and South Asia were vulnerable to drought, while North America and Europe were less vulnerable to drought. We analyzed the global spatial distribution of drought frequency, and we found that socio-economic drought-prone areas in GDIS were generally consistent with hydro-meteorological drought-prone areas expressed by ERA5 Land based soil moisture deficit (the Horn of Africa, northern China, and western India). The use of sub-national disaster information, such as GDIS, makes it possible to identify socio-economic drought-prone areas on a finer scale and can contribute to validating reanalysis products.

**Code and data availability**

The drought clustering python code can be downloaded at https://github.com/julherest/drought_clusters (last access: 5 August 2022). The ERA5-Land dataset can be downloaded at https://cds.climate.copernicus.eu/cdsapp#!/dataset/reanalysis-

era5-land?tab=form (last access: 5 August 2022). The GDIS dataset can be downloaded at https://cmr.earthdata.nasa.gov/search/concepts/C2022273992-SEDAC.html (last access: 5 August 2022). The EM-DAT database can be viewed at https://www.emdat.be/ (last access: 5 August 2022). World bank's geographical regions can be viewed at https://datatopics.worldbank.org/world-development-indicators/the-world-by-income-and-region.html (last access: 5 August 2022). The MODIS land cover data can be downloaded at https://lpdaac.usgs.gov/products/mcd12c1v006/ (last access: 5 August 2022). The gridded population of the world can be downloaded at https://sedac.ciesin.columbia.edu/data/set/gpw-v4-population-density-rev11 (last access: 5 August 2022). The global map of FAO's annual average precipitation can be viewed at World bank website https://data.worldbank.org/indicator/ag.lnd.prcp.mm?msclkid=215b9959b08711ec944832810373c8aa&view=map (last access: 5 August 2022).

**Author contribution**

Y.S. and Y.K. designed the experiments and Y.K. carried them out. Y.K. prepared the manuscript with contributions from Y.S. Y.S. acquired the funding.

**Competing interests**

The authors declare that they have no conflict of interest.

**Acknowledgements**

We thank Jakob Zscheischler and Mariana Madruga de Brito for their constructive comments. This work was supported by JAXA grant (ER2GWF102 and ER3AMF106), JSPS KAKENHI grant (21H01430), and Katsu Kimura Research Award.

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

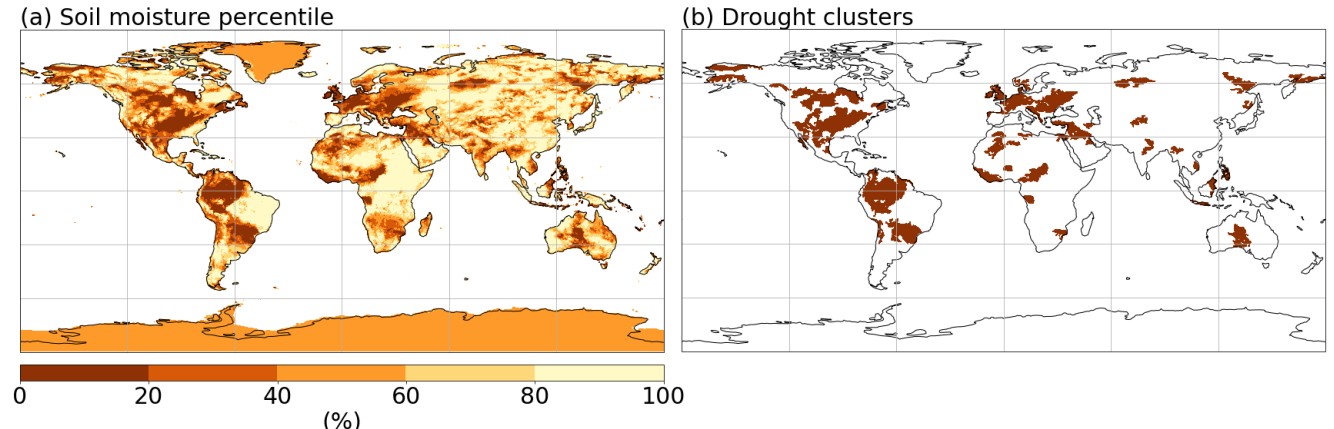

**Figure 1: Demonstration of drought clustering. (a) Global map of soil moisture percentile for the root-zone layer's soil moisture in January 1964. (b) Drought clusters, spatially contiguous areas under drought (below 20th percentile) are extracted from (a). A 2-D median filter is applied prior to drought clustering, which makes slight differences compared with (a).**

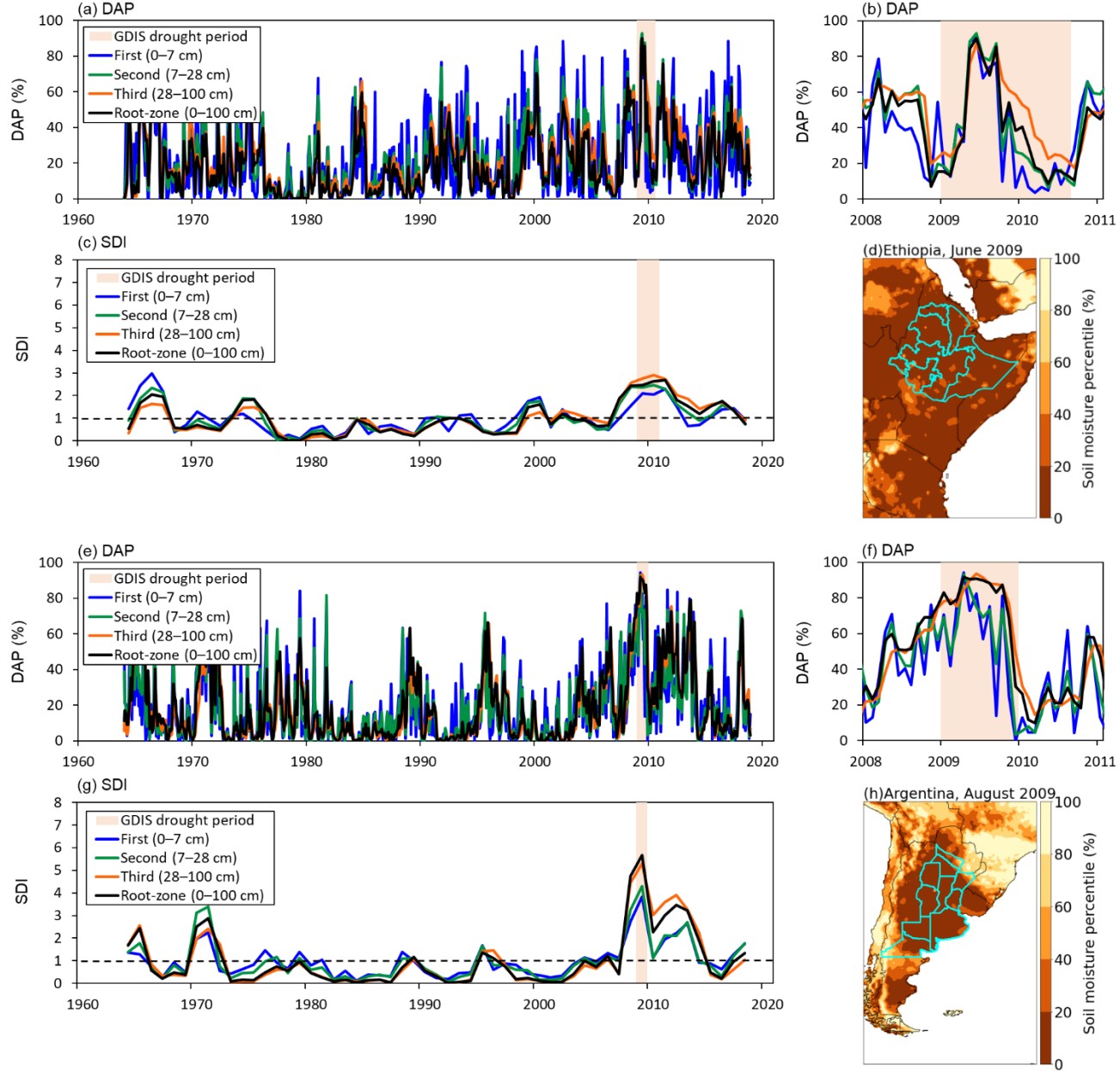

**Figure 2: Demonstration of drought indices in different soil layers for the drought events in Ethiopia in 2009 (a–d) and Argentina in 2009 (e–h). (a, e) DAP; red band shows the GDIS drought period, and each coloured line shows the values in each soil layer (first (blue), second (green), third (orange), and root-zone (black)). (b, f) the enlarged view of DAP around the GDIS drought period. (c, g) SDI; the legends are the same as DAP (a, e), and grey dotted line shows the value of 1 (the mean of SDI over the study period). (d, h) the GDIS drought area of this event; black line shows the country border, and light blue line shows the affected administrative units shown in GDIS. GDIS provides GIS polygons of administrative units, and administrative units with the same event identifier (*disasterno*) were treated as one "GDIS event area", the assembles of each light-blued administrative unit. The soil moisture percentile is generated from the root-zone layer's soil moisture as an example.**

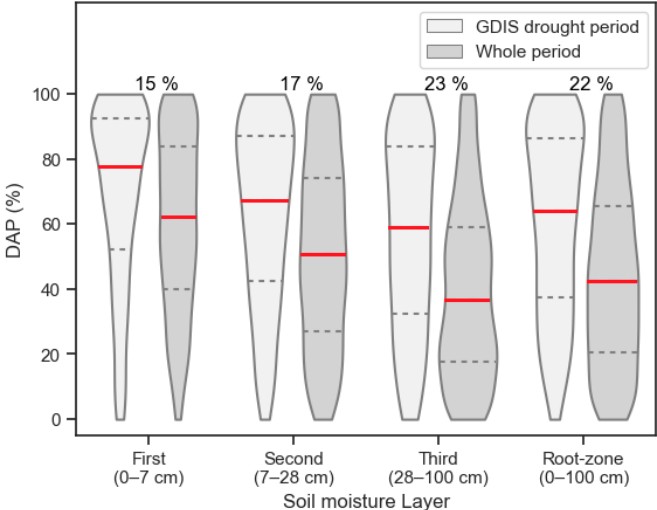

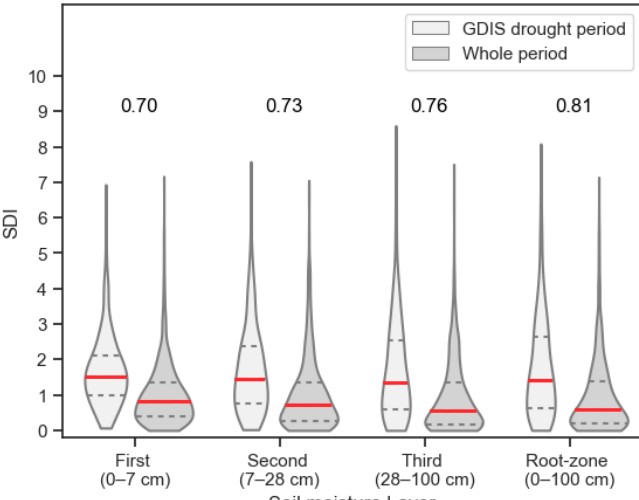

Figure 3: Comparison of DAP between the GDIS drought period and the whole period. The red line shows the median value and grey dotted lines show the 25th and 75th percentile values of each distribution. The value above the violin plot shows the difference of the median values in the GDIS drought period and the whole period.

Figure 4: Comparison of SDI between the GDIS drought period and the whole period. The red line shows the median value and grey dotted lines show the 25th and 75th percentile values of each distribution. The value above the violin plot shows the difference of the median values in the GDIS drought period and the whole period.

620

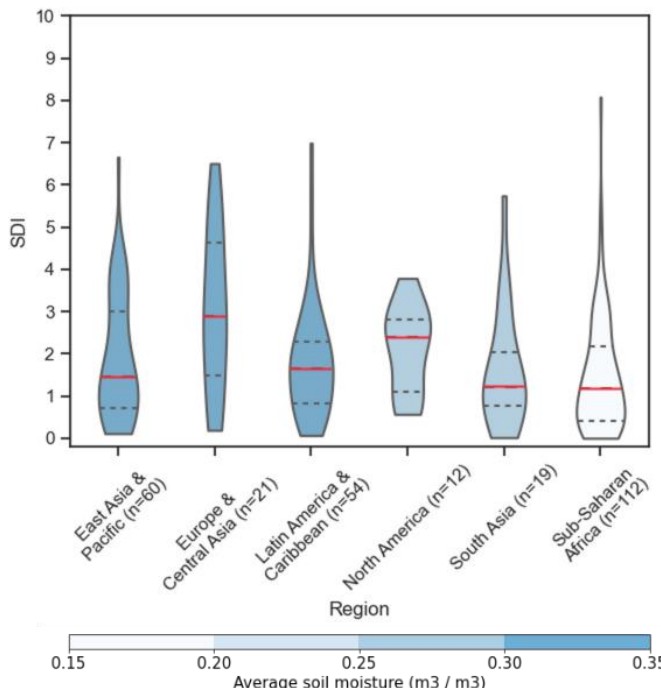

**Figure 5: Comparison of the root-zone SDI by geographical regions. The red line shows the median value and grey dotted lines show the 25th and 75th percentile values of each distribution. The colour shows the average soil moisture over the study period (1964– 2018).**

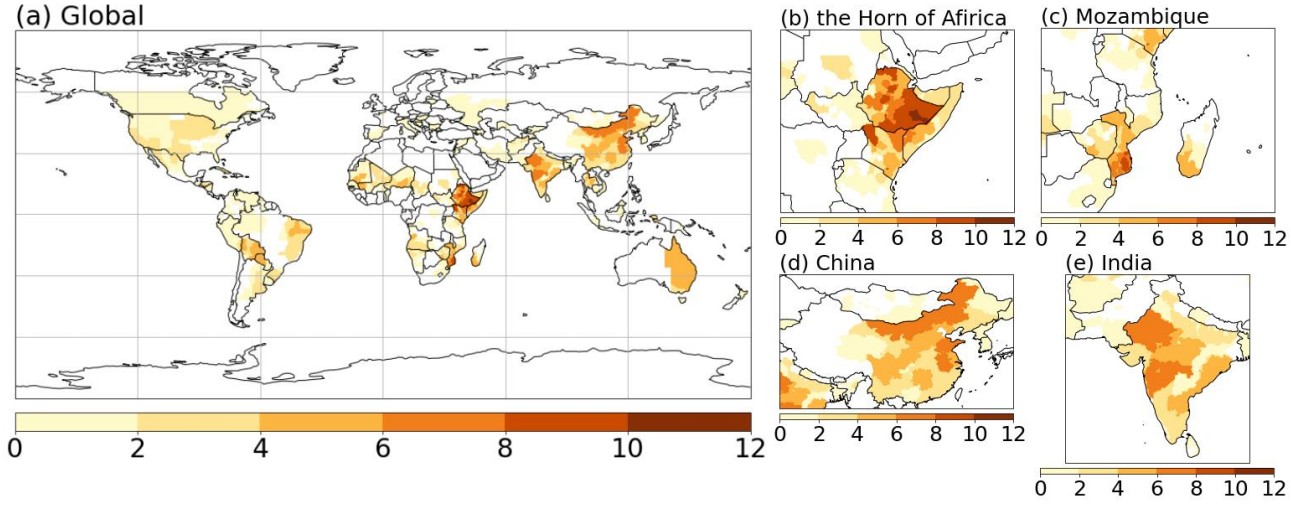

**Figure 6: The number of drought events based on GDIS. (a) Global map, enlarged view of (b) the Horn of Africa, (c) Mozambique, (d) China, and (e) India.**

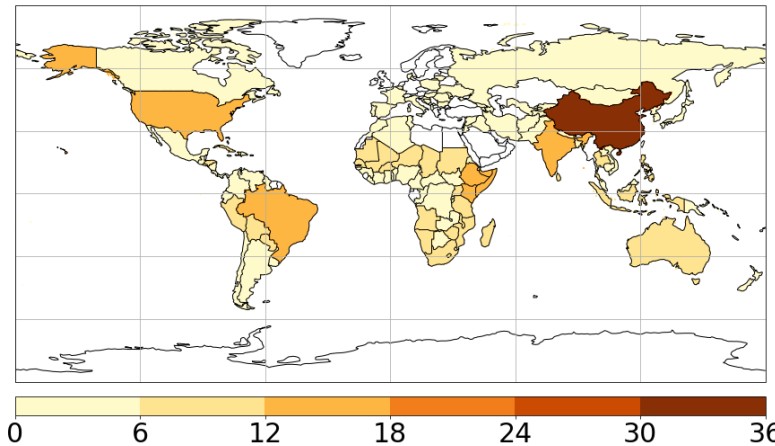

**Figure 7: The number of drought events based on EM-DAT.**

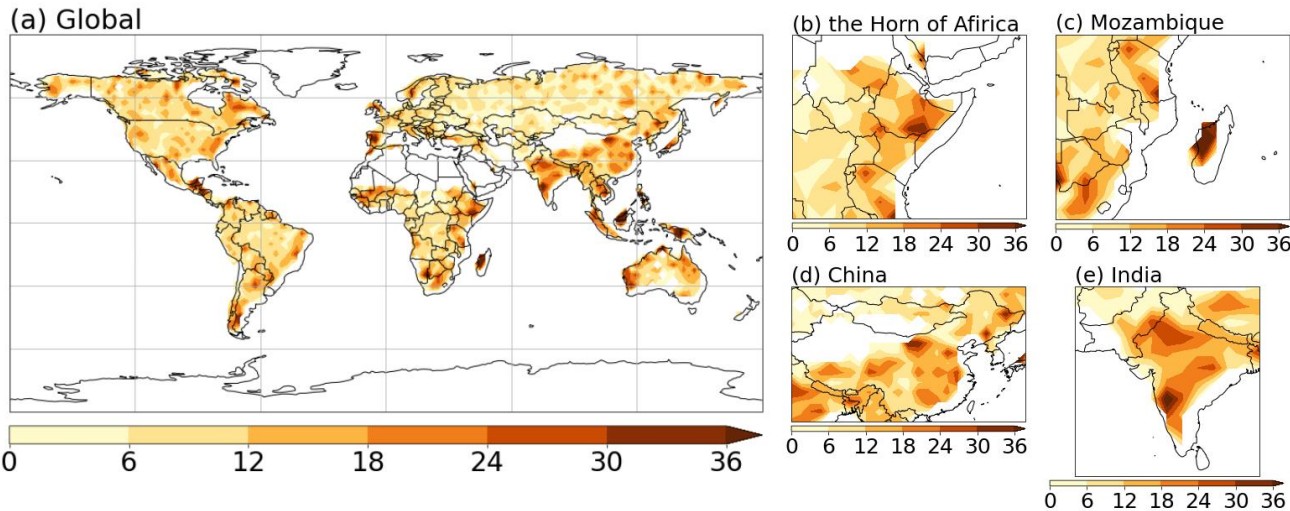

**Figure 8: The number of drought cluster centroids based on ERA5-Land. (a) Global map, enlarged view of (b) the Horn of Africa, (c) Mozambique, (d) China, and (e) India.**