# Peer review of "Global assessment of sub-national drought impact based on the Geocoded Disasters dataset and land reanalysis"

_Hydrology and Earth System Sciences, 2022_

## Author Comment (AC1)

Response letter of hess-2022-124-RC1

Dear Jakob Zscheischler,

Please find the responses to the comments.

Comments made by the reviewer were highly insightful. They allowed us to greatly improve the quality of the manuscript. We described the response to the comments.

Each comment made by the reviewers is written in italic font. We numbered each comment as (n.m) in which n is the reviewer number and m is the comment number.

Sincerely,

Yuya Kageyama and Yohei Sawada

**Responses to the comments of Referee #1**

*General comments*

*This a valuable contribution as it links drought entries in a recently constructed disaster database (GDIS), which is based on EM-DAT, with actual droughts identified based on a state-of-the-art reanalysis dataset. The work is important because typically disaster databases are not based on information derived from meteorological variables and establishing such links gives credence to the databases but also serves to identify hazard thresholds and differences in vulnerability.*

*Generally the manuscript is very clear and easy to follow but it would benefit from a more in-depth discussion of a number of points that are only briefly mentioned. More detailed comments and suggestions follow below.*
→ Many thanks for the comments and suggestions.

*(1.1) L10: "We found that ERA5-Land soil moisture accurately captured the socio-economic impacts of drought shown in GDIS." I think I understand what the authors mean but as it is this statement is not correct. ERA5-Land cannot capture "impacts of droughts", rather it provides information that droughts actually occurred when disasters labelled as drought-driven where recorded. Similar statements can be found in other places throughout the text. Please adjust the language to more correctly represent your findings and the relationship between drivers of disasters and the recorded impacts.*
→ We will change the sentences like "We found that the socio-economic impacts of drought shown in GDIS were generally represented by drought hazards quantified from ERA5-Land soil moisture."

*(1.2) L12: "were robust": better us "were less vulnerable"*
→ We appreciate this comment. We will replace "were robust" with "were less vulnerable."

*(1.3) L106: The authors specify some criteria for the analysed events but it seems the number of analysed events is not affected by this. In line 100 it says "The 282 drought events…were analysed", which suggests that this is the number of events contained in GDIS. In Line 162 it is stated that there are 282 drought events. Please clarify how many events are available in GDIS and how many are finally analysed in this study.*
→ Originally, there are 433 drought events in GDIS and 282 drought events were analyzed. Therefore, the number of analyzed events was indeed affected by our quality control. This point was indeed unclear in the original version of the paper. We will add this information in the revised

version of the paper.

(1.4) *L 163: "We recognized that the drought indices successfully capture the GDIS drought events if the two distributions are not statistically the same." This is only true for the full distribution, not at the individual event level. There could still be many events where no drought is evident from a soil moisture perspective (there is also a strong overlap in the distributions). Furthermore, the test doesn't tell you how the distributions differ. Theoretically it would be possible that all disaster events show less extreme drought conditions compared to the control and the KS test give significant results. Please adjust this statement accordingly and make it more nuanced. This should also be reflected in the abstract.*

→ We fully agree with this comment. In the results section 4.1 (L 208, L 212), we stated that the drought index during the GDIS drought periods was significantly higher than that of the whole period. We checked if the distribution is high/low, as well as the results of *K–S* test.

We will change "We recognized that the drought indices successfully capture the GDIS drought events if the two distributions are not statistically the same." to "We recognized that the GDIS drought events are generally represented by the drought indices quantified from ERA5-Land if the median of the drought index during the GDIS drought periods is higher than that of the whole period and the two distributions of the drought index are not statistically the same." In the results section (from L 217), we will add "Please note that although we confirmed a general linkage between drought hazards and the GDIS drought events, some GDIS events were not explained from a soil moisture perspective."

(1.5) *L 165: unclear what "socio-economic drought events in GDIS" means. I assume "socio-economic droughts" are the droughts events and impacts recodred in GDIS? Better use something like "drought disasters" or similar*

→ As the reviewer mentions, "socio-economic droughts" are the drought events and impacts recorded in GDIS. We will replace "socio-economic drought events in GDIS" with "drought events in GDIS", to be consistent with the expressions in other places of our manuscript.

(1.6) *L170: Initially I wasn't completely sure what the purpose of the drought clustering is. I assume the idea is to check whether spatially large droughts typically also lead to impacts (as recorded by EMDAT/GDIS). Given that the drought definition is percentile-based, every location experiences drought with the same frequency, and differences from a rather homogeneous distribution (as for instance visible in the US, Canada and Russia in Fig. 8) are driven by the distribution of continents and the choice of the spatial cutoff (100000km2). With this background, this could be a useful*

*analysis but it would be good to motivate it better and discuss the results in more detail (e.g. it seems that some regions experience more contiguous/large-scale droughts than others, why?). The finding that drought disasters tend to occur in regions that are characterise by frequent large-scale droughts is then quite interesting and novel, especially because from a meteorological perspective and at a pixel level, the frequency of drought occurrence is the same everywhere (20% of the time in this study). So it seems that drought disasters occur when droughts occur over large areas. These findings could be described and discussed in more detail.*

→ As the reviewer mentions, the idea of drought clustering is to check whether spatially large hydro-meteorological droughts typically lead to disasters that can be seen in GDIS. In the revised version of the paper, we will clarify the role of drought clustering more by the following sentences. "The consistency between hydro-meteorological drought-prone areas in ERA5-Land and socio-economic drought-prone areas in GDIS shows that spatially large hydro-meteorological droughts (we analyzed at least 100,000 km$^2$) typically lead to impacts as shown in GDIS. Although the drought frequency defined by simulated soil moisture is the same everywhere at the grid level (we set the 20th percentile as a drought threshold), there was considerable heterogeneity in the spatially large drought-prone areas (Fig. 8)."

To reinforce our idea, a sensitivity analysis with different thresholds of the size of drought clusters will be described in the supplement material (see the results in the lowermost part of this response). We have already found that drought-prone areas found in GDIS cannot be reproduced by ERA5-based drought-prone areas when we used too small or large thresholds. We will mention it in the revised version of the paper by describing "See also the supplement material for a sensitivity analysis with different thresholds of the size of drought clusters (Fig. S3), showing that drought-prone areas found in GDIS cannot be reproduced by ERA5-based drought-prone areas when we used too small or large thresholds that are used to identify drought clusters."

We will also discuss the results more deeply focusing on the underlying climatological mechanisms which are useful to interpret the drought-prone areas. In the revised version of the paper, we will add, "There are some factors that contribute to the emergence of drought-prone areas, such as El Niño-Southern Oscillation (ENSO), La Niña, intertropical convergence zone (ITCZ), monsoon, land-atmosphere coupling, and anticyclones (Christian et al., 2021). La Niña affects the Horn of Africa, northern China, and western India and has caused severe drought impacts (Funk, 2011; Jain et al., 2021). Ummenhofer et al. (2011) clarified the effect of El Niño–Indian monsoon relationship on drought in western India. Quite large spatio-temporal events such as La Niña might cause drought to persist, which leads to drought impacts as shown in GDIS. However, the drought factors are complex, and much future work is needed to reveal the mechanism of the emergence of droughtprone areas. "

These two points (i.e., the role of drought clustering and climatological mechanisms) will be included in the discussion section of the revised paper.

**Supplement material**

**S3 Sensitivity of drought cluster centroids map to the size of drought clusters**

Upscaled map of the drought cluster centroids was generated using four thresholds of the size of drought clusters: 10,000km$^2$, 50,000km$^2$, 100,000km$^2$, and 500,000km$^2$, shown in Fig. S3. The drought clusters were generated from the third layer's soil moisture. In the results part, we showed the case of 100,000km$^2$. This shows that drought-prone areas identified by drought indices are not well consistent with those found in GDIS, whether the size of drought clusters is too large or too small.

[Figure]

**Figure S3: Sensitivity of drought cluster centroids map to the size of drought clusters. (a) 10,000km$^2$, (b) 50,000km$^2$, (c) 100,000km$^2$, and (d) 500,000km$^2$.**

(1.7) *Another point to discuss is that you're using a relative percentile, which means that in generally wet regions, a drought de fined in this way might not be that impactful because the absolute amount of available water is still quite high. This will to some extent also determine where drought disasters occur and might confound the vulnerability assessment. It makes sense to use a relative percentile given that ecosystems/societies are usually adapted to the water availability in their region. However, it is worth discussing this choice.*

→ We appreciate this comment. The major limitation of using an absolute value of soil moisture is that it is difficult to conduct a unified drought analysis across multiple regions. As the reviewer mention, the thresholds of drought impact occurrences are different in different regions because ecosystems/societies are usually adapted to the water availability in their region. Another point is the uncertainty of using data with different reanalysis products or different observation locations. Many climate studies have used relative values rather than absolute values, for which climate model biases are less important (Liu and Key, 2016). Palecki (2018) also stated the uncertainties of absolute soil moisture product from different soil moisture observation networks with different measurement equipment and calibrations.

The limitation of a relative percentile-based drought quantification is that the drought in extreme wet regions might not be represented as the reviewer points out. In the drought clustering analysis, we showed that Indonesia was hydro-meteorological drought-prone areas in ERA5-Land, which were not included in drought-prone areas found in GDIS. We attributed this inconsistency to the abundance of absolute water availability in Indonesia (from L287 to L290).

The issues discussed above will be included in the revised version of the paper: "We used the percentile soil moisture, deviation from the normal condition, to quantify drought following many previous studies (e.g., Sheffield and Wood, 2011; Hanel et al., 2018). However, the inconsistency for Indonesia between hydro-meteorological drought-prone areas and drought-prone areas found in GDIS implies that the drought in extreme wet regions might not be well represented. It implies that our drought quantification method based on relative values of soil moisture cannot accurately consider the amount of regularly available water resources. An alternative way to quantify drought is to use an absolute soil moisture value, but it is not straightforward to quantify drought events by absolute soil moisture values. The thresholds of drought impact occurrences in absolute values are different in different regions, because ecosystems/societies are usually adapted to the water availability in their region. This means that a unified drought analysis across multiple regions is difficult. Another limitation of the absolute value is the uncertainty of using data with different reanalysis products or different observations. Many climate studies have used relative values rather than absolute values, for which climate model biases are less important (Liu and Key, 2016). Palecki (2018) stated the uncertainties of absolute soil moisture product from different soil moisture observation networks with different measurement equipment and calibrations."

(1.8) *L206: You could mention some more details in the results. For instance, how large are the differences between the medians for the different soil moisture layers in Figs 3 and 4. I assume the SDI for the whole period is approx 1 by definition (Fig. 4)? I would also mention that for*

*clarification.*

→ We will describe how large are the differences. The mean value of SDI for the whole period is approximately 1 by the definition (Fig. 4).

(1.9) *L219: It maybe worth checking how these regions differ in their absolute SM values (averaged over time). Regions with lower water availability in general might experience drought disasters more frequently even though the relative deviation from normal conditions is small (see the comment higher up). Independent of the findings, the interpretation that the identified regions are more vulnerable to drought probably still holds, but I think it can slightly change the interpretation and consequences for resilience planning. If water is typically abundant, it's much easer to be drought resistant.*

→ We appreciate this implication. We will replace Fig. 5 with the one shown below, whose colors of the violins show the averaged absolute soil moisture value in each region. Sub-Saharan Africa, which was vulnerable to drought, shows lower water availability. This may be one of the reasons for the difficulty for Sub-Saharan Africa to manage the drought hazards. We will add this point in the results and discussion sections of the revised paper.

[Figure]

**Figure 5: Comparison of the root-zone SDI by geographical regions. The red line shows the median value and grey dotted lines show the 25th and 75th percentile values of each distribution. The color shows the average soil moisture over the study period (1964–2018).**

(1.10) *L258: Note that the results of Fig. 5 also confirm results from Tschumi & Zscheischler (2020)*

*who also found smaller climate anomalies in less developed countries during EMDAT disasters for different climate variables (their Fig. 9).*

→ We appreciate this comment. We will discuss that our result is consistent with this work in the revised version of the paper.

(1.11) *L280: unclear what "reproducibility of ERA5-Land" means. Please clarify.*

→ "Reproducibility of ERA5-Land" means the performance of ERA5-Land to simulate soil moisture. We will rephrase "The lack of the reproducibility of ERA5-Land might affect these inconsistencies." to "The performance of ERA5-Land to simulate soil moisture might affect these inconsistencies."

(1.12) *L295: again, unclear what "reproducibility" of reanalysis products means. It seams the authors mean that using different climate datasets, one obtains similar drought maps (which makes sense) whereas this does not hold for socioeconomic impacts. This of course depends on how these things are defined. If the same drought definition is used, the choice of (climate) dataset has only little effect on the analysis as this information is comparably well constrained by observations. For socio-economic impacts, no clear/objective definition of disasters etc. exists and uncertainties in impact estimates are usually high. So a comparison across datasets is difficult also because different datasets use different definitions.*

→ The "reproducibility" (L295) means the performance of reanalysis products to simulate hydrometeorological variables. We believe that the validation of the performance has not been fully conducted in terms of whether the anomalies of simulated variables are consistent with the socio-economic impact of drought events in disaster databases. This "reproducibility" does not imply the similarity of results with different reanalysis products. We believe that this comment is provided because we failed to deliver this point using the misleading word of "reproducibility". In the revised version of the paper, this wording is no longer used. We will change this section (from L 295) to, "Although various reanalysis products have been developed and their validations have been conducted by comparing them with earth observation data (e.g., Muñoz-Sabater et al., 2021; Reichle et al., 2017; Rodell et al., 2004), few studies have examined the validation in terms of the disaster occurrence. Sawada (2018) compared the areas identified as drought quantified from a reanalysis product with the disaster records from EM-DAT, but only in a country-scale. As seen in Fig. 7, national-level information does not provide accurate views of disaster locations, so that it is insufficient for validation data. The use of sub-national disaster databases such as GDIS opens the door to validate reanalysis products in terms of the disaster occurrence."

(1.13) *L328: "GDIS only covers about 60% of droughts in EM-DAT" Can you comment on why this*

*is the case? Was more detailed information on the remaining events not available?*

→ This is due to vague or unknown location names in EM-DAT. We will add this point in the revised version of the paper.

Additional References

Christian, J., I., Basara, J. B., Hunt, E. D., Otkin, J. A., Furtado, J. C., Mishra, V., Xiao, X., and Randall, R. M.: Global distribution, trends, and drivers of flash drought occurrence, Nat. Commun., 12, https://doi.org/10.1038/s41467-021-26692-z, 2021.

Funk, C.: We thought trouble was coming, Nature, 476, 7, https://doi.org/10.1038/4760007a, 2011.

Jain, S., Mishra, S. K., Anand, A., Salunke, P., and Fasullo, J. T.: Historical and projected low-frequency variability in the Somali Jet and Indian Summer Monsoon, Clim. Dynam., 56, 749–765, https://doi.org/10.1007/s00382-020-05492-z, 2021.

Liu, Y. and Key, J. R.: Assessment of Arctic Cloud Cover Anomalies in Atmospheric Reanalysis Products Using Satellite Data, J. Climate, 29, 6065–6083, https://doi.org/10.1175/JCLI-D-15-0861.1, 2016.

Palecki, M. A.: Using Standardized Soil Moisture Indices for Drought Monitoring, in: AGU Fall Meeting Abstracts, H53J-1725, 2018.

Ummenhofer, C. C., Sen Gupta, A., Li, Y., Taschetto, A. S., and England, M. H.: Multi-decadal modulation of the El Nino-Indian monsoon relationship by Indian Ocean variability, Environ. Res. Lett., 6, https://doi.org/10.1088/1748-9326/6/3/034006, 2011.

---

## Author Comment (AC2)

Response letter of hess-2022-124-RC2

Dear Mariana Madruga de Brito,
Please find the responses to the comments.
Comments made by the reviewer were highly insightful. They allowed us to greatly improve the quality of the manuscript. We described the response to the comments.

Each comment made by the reviewers is written in italic font. We numbered each comment as (n.m) in which n is the reviewer number and m is the comment number.

Sincerely,
Yuya Kageyama and Yohei Sawada

**Responses to the comments of Referee #2**

*In this manuscript, the authors use the new GDIS data and link it to hazards (i.e. soil moisture). The contribution is timely and innovative. I enjoyed reading the paper. Currently, we have little understanding of the relationships between impacts and hazard data. Hence, the proposed approach provides a way to understand these linkages. I particularly appreciate that instead of the traditional approach (e.g. hazard x vulnerability x exposure = risk), the authors go from the disaster itself, and from it try to identify the hazard drivers.*

*The manuscript is well written, yet, some issues need to be addressed:*

→ Many thanks for the comments and suggestions.

(2.1) *My main criticism is that the authors claim to have assessed the "regional vulnerability to drought" (see line 258). Figure 5 shows that the SDI alone cannot explain drought disasters. Indeed, in Sub-Saharan Africa and South Asia, the SDI is not as high as in Europe and North America, yet, many disasters are observed. One explanation for this is indeed, the socio-economic vulnerability in these regions is higher. However, it could also be that it is the population exposure. Since you have not accessed any of these, you cannot make claims regarding the vulnerability.*

→ We appreciate this comment. As the reviewer mentions, we did not directly include exposure, and "regional vulnerability to drought" was overstated. There are several difficulties in treating exposure in our study. To estimate exposure, it is necessary to identify what has been damaged (e.g., people, crops, forests, etc.). However, EM-DAT provides no information on impact types (e.g., water shortage, famine, wildfire, etc.) in many drought events, which inhibits the identification of what has been damaged. In addition, EM-DAT provides no information about the amount of damage in multiple drought events (line 329). Therefore, we will change "regional vulnerability to drought" to "proxy of regional vulnerability to drought". In addition, we will explicitly say that our analysis did not directly include exposure in the discussion (See also our response to the comment 2.7). In the abstract (line 11), we will change "Our comparison between GDIS and ERA5-Land can quantify vulnerability to drought" to "Our comparison between GDIS and ERA5-Land could benefit the quantification of vulnerability to drought."

(2.2) *You mentioned that you disregarded drought events that lasted for less than 2 months. However, I assume most of the EM-DAT data only has a year, and few have a start and end date. In such cases, you mentioned that you considered the whole year as a drought year. That can be very problematic, as probably many of these events could actually be 2 months. It would be great if you could bring this limitation in the discussion section, where you already mentioned the limitation of the impact data.*

→ We fully agree with this comment. There are some GDIS drought events in which the details of drought duration information are not provided. Therefore, drought events shorter than two months may be included in our analysis, although we intended to exclude them. We will add this limitation in the discussion section from line 331. "Although we excluded GDIS drought events shorter than two months from our analysis, some of the analyzed events might be shorter than two months. This is because we applied January for the start and December for the end of the event if the start and/or end months of events shown in EM-DAT were unclear."

(2.3) *Why was landcover data from 2020 used (line 127) if the drought events go from 1964 to 2018? Likewise, why did you consider soil moisture data from 2019-2020 (line 132) when your impact data is from previous years?*

→ The detailed landcover datasets such as MODIS landcover (used in our study), GlobCover, and Global Land Cover Characterization (GLCC) do not provide landcover data over a long period of time. The MODIS land cover dataset starts in 2001. In addition, we only used landcover data to exclude the barren or sparsely vegetated areas from drought-prone areas by the drought clustering method, so that the timing of land cover does not significantly change our results. We confirmed that there was little or no difference in the extent of the barren or sparsely

vegetated areas when we compared with the oldest 2001 data and the latest 2020 data. We will not change the manuscript at this point.

We used (absolute) soil moisture data during 1950–2020 in calculating the percentiles. After the percentiles were calculated, only (percentile) data during the period of the GDIS drought events (1964–2018) was used (line 131). We expected that a longer data period would contribute to more robust percentile values. For calculating SPI, at least 30 years of data is required (McKee et al., 1993). We also used percentile values which were calculated from absolute values during 1964–2018 (the same period with the GDIS drought events) and the results were slightly better when used percentile values which were calculated from absolute values during 1950–2020; the differences of drought indices during the GDIS drought period and the whole period was bigger and hydro-meteorological drought-prone areas were more consistent with GDIS drought-prone areas (not shown). We will add a comment from line 133. "We used the longer period of original ERA5-Land data (1950–2020) than the study period (1964–2018) to yield more robust percentile values."

(2.4) *Item 3.3 and 4.2. Similar to the first comment, stratifying your SDI data by different regions is not equivalent to "understanding the vulnerability in each region". What you did here was to analyze the SDI according to different regions. This is definitely an important analysis, but it is not a vulnerability analysis. You can hypothesise that the vulnerability causes these differences, but you cannot confirm this with the present data. The statements in line 255 are pretty strong and cannot be made without considering the socio-economic vulnerability. I suggest toning them down and writing that "the vulnerability could explain these differences". However, for this, further analysis is required.*

→ We appreciate this comment. Relating to the first comment (2.1), we will tone down like "vulnerability could explain these differences." We will also change the subtitle "Regional vulnerability to drought" to "Regional levels of drought indices associated with GDIS drought events". The detailed comment on exposure will be described in the discussion section (See also comment 2.7). Below is the revised version of these sections.

**3.3 Regional levels of drought indices associated with GDIS drought events**

The levels of hydro-meteorological drought indices associated with socio-economic drought events identified in GDIS are different in different regions. Vulnerability could explain these differences (Delbiso et al. 2017; Gasparrini et al. 2015; Tschumi and Zscheischler, 2020) Please note that vulnerability is not the only explanation for these differences; exposure is another factor that influences the linkage between hazards and impact (Visser et al., 2014; see also the discussion section). Since we did not directly include exposure, we recognized these differences as "the proxy of vulnerability". Following Bachmair et al. (2016), the levels of SDI which are associated with socio-economic drought events in GDIS were quantified and analyzed. The levels of SDI were stratified by geographical regions to understand the distribution of the proxy of vulnerability in each region.

**4.2 Regional levels of drought indices associated with GDIS drought events**

Figure 5 shows ~. A large number of small SDI events indicates that less severe hydro-meteorological droughts have caused serious socio-economic impacts, meaning that the regions are vulnerable to drought. On the other hand, the regions with a large number of large SDI events can be recognized as robust regions to drought. Thus, Sub-Saharan Africa and South Asia are vulnerable to drought, while North America and Europe are less vulnerable to drought. This regional characteristic of vulnerability to drought can be found when SDI is generated by soil moisture in different soil layers (not shown). Note that Middle East & North Africa were excluded from the analysis because the sample size was too small (n = 4).

In the introduction (line 72), we will change "we quantified the vulnerability to drought in different geographical regions" to "we quantified the levels of drought indices associated with GDIS drought events in different geographical regions, whose differences could benefit the quantification of vulnerability to drought".

(2.5) *Item 3.4. It is not clear to me what the advantage of clustering is. Why would the results be different with or without clustering? Could you please elaborate?*

→ The idea of drought clustering is to check whether spatially large hydro-meteorological droughts typically lead to disasters that can be seen in GDIS. For more detail, please see our responses to the comment of the reviewer #1 (comment 1.6).

(2.6) *In line 259, you say that your findings are consistent with previous ones that used GDP, however, you have not used GDP here. Please rephrase the sentence, or do the analysis*

→ We intended to say that high-income countries were associated with lower vulnerability to drought. We did not use GDP, as the reviewer mentions. We will change this sentence like "Previous studies have shown that higher GDP per capita is associated with lower vulnerability to natural hazards (e.g., Kim et al., 2019; Tanoue et al., 2016). North America and Europe are high-income countries, and these previous works support our findings."

(2.7) *In line 266, you write that the vulnerability is explained by the lack of structural measures. Sure, these are contributing factors, but socio-economic variables and exposure could be even more important. I understand the author's point of view, being an engineer myself. However, "command and control" approaches in which drought risk is controlled solely with conventional engineering measures is an outdated view. The authors cannot ignore the vast literature on socio-economic vulnerability and risk perception. You do not need to cite these papers, but have a look at the GAR chapters on vulnerability and droughts:*

*https://gar.undrr.org/sites/default/files/chapter/2019-05/Chapter_3.pdf*

*https://gar.undrr.org/chapters/chapter-6-special-section-drought*

→ We appreciate this implication. We stated the complex nature of vulnerability in the discussion sections (from line 319) and included non-structural countermeasures. We will strengthen the content including exposure in this section (from line 323), "Exposure is another important factor that influences the linkage between hazards and impact (Visser et al., 2014). The inconsistency between hydro-meteorological drought-prone areas and drought-prone areas found in GDIS in Namibia implies that the quantification of exposure is necessary to strengthen the analysis in our study. It is necessary to identify what has been damaged (e.g., people, crops, forests, etc.) to quantify exposure. However, EM-DAT provides no information on impact types (e.g., water shortage, famine, wildfire, etc.) in many drought events, which inhibits the identification of what has been damaged. Like vulnerability, exposure is complex and dynamic, for example, affected crops change with the seasons (Bodner et al., 2015). To improve our analysis on vulnerability shown in Section 4.2, detailed analyses on the complex and dynamic nature of both vulnerability and exposure are necessary."

(2.8) *Line 282: Spain is not shown in Figure 5. Hence, you cannot make this statement based only in the Figure. It could well be that the other European countries are "robust" and push the mean up, whereas Spain pushes the mean down. I suggest removing the sentence or doing the analysis for Spain only.*

→ Spain is actually less vulnerable to drought. We will add the mean value of SDI for Spain in the manuscript.

(2.9) *Line 301: I find it good that you can find good linear correlations. However, I was surprised as soil moisture and drought impacts usually do not have a linear relationship. Indeed, there are non-linear relationships (there is*

*a lag between soil moisture and forestry impacts for instance). Furthermore, you have multi-year drought events, which add complexity to the interpretation of spatial and temporal differences in drought prevalence. Indeed, it could be that a drought is not so severe in biophysical terms. However, because they last so long, the impacts are high. I assume you got good correlations because EMDAT usually do not have long term impacts but rather focuses on immediate ones. Please add a discussion of these issues in your discussion section.*

→ We fully agree that the relationship between soil moisture and drought processes is non-linear. As the reviewer mentions, there are non-linear relationships between the drought severity and drought stress in vegetation (e.g., Chen et al. 2020; Meyer et al., 2014), where damage increases suddenly when the drought severity exceeds a certain critical threshold. Note that we found no "good linear correlations", nor did we explore any linear relationships in this paper. We did not explicitly describe that our analyzed relationship is linear. It is currently unclear for us which part of our analyses the reviewer recognized as a good linear correlation. Maybe we can improve our responses to this comment if the reviewer clarifies this point in the next round.

We agree that there is a lag between hazards and impact. Whether EM-DAT captures the long-lasting drought impacts is an important issue. The reason why drought events shown in GDIS were generally represented by drought hazards quantified from ERA5-Land may be due to the relatively long period of the drought duration in GDIS; the mean duration is approximately 12 months. Anyway, drought phenomenon is complex; the impacts of drought last even after the hydro-meteorological drought ends. Further analysis is needed to focus on the chronological correspondence to drought hazards. We will add the following discussion from line 318 emphasizing the complex nature of the relationship between soil moisture and drought impacts.

"The relationship between hazards and impact is much more complex than addressed in this study. Many studies have revealed the non-linear relationships between the drought severity and the reduction of vegetation growth (e.g., Chen et al. 2020; Meyer et al., 2014), where damage increases suddenly when the drought severity exceeds a certain critical threshold. On the other hand, de Brito et al. (2020) reported that there was a linear relationship between the drought severity and the number of drought articles as a proxy of socio-economic drought impacts. In addition, drought is a long-lasting disaster and there is a time-lag between hazards and impact, so that the period of hydro-meteorological drought is not necessarily consistent with the period considered as a disaster in EM-DAT. Some studies have revealed that the impacts of drought last even after the hydro-meteorological drought ends (e.g., Shahbazbegian and Bagheri, 2010). Yokomatsu et al. (2020) tried to capture the impact of the drought in terms of the economic development after the drought. In any case, further analyses are needed to focus on the chronological correspondence to drought hazards."

*(2.10) Line 9: What is meant by "drought information" here? Do you mean Drought impact information (i.e., the GDIS data?). Please be more specific.*

→ This is drought impact information as shown in GDIS. We will change the language.

*(2.11) Line 55-57: I do not understand the purpose of these 3 last sentences. When you write about vegetation growth, do you mean forest or crops? Regarding the last sentence. Why do we need to treat drought impacts as they are socially perceived? If you think the section is relevant, please expand it to make it clearer to the reader.*

→ Vegetation means plants in general, both forests and crops. We intended to say that not all droughts affect vegetation growth, and not all vegetation decline is caused by drought. "Socially perceived as drought" implies that we should consult a disaster database, which shows events in which the society has actually suffered from drought (we say it as "socially perceived"). We will rephrase like, "It is unclear whether socio-economic drought impacts are associated with declined vegetation growth. It is ideal to treat the socio-economic drought impact

based on a disaster database since it directly shows events in which the society has actually suffered from drought."

(2.12) *Line 95: This information is repeated in the introduction. I suggest removing it: "GDIS is the geocoded…."*

The next sentence is about EM-DAT, which made us say "GDIS is the geocoded disaster locations database based on EM-DAT". We will rephrase like "GDIS is generated based on EM-DAT."

(2.13) *Line 122: You do not "show the drought vulnerability". Land cover is an exposure data, not vulnerability.*

→ We will change "show the drought vulnerability" to "show the levels of drought indices associated with GDIS drought events."

We intended to use land cover data as exposure data. We will change it like "As a proxy of exposure data, we used the MODIS land cover~."

(2.14) *Line 207: "significantly higher": could you please provide the test results that made you reach this conclusion.*

→ The test results are below the table. In any case, the values are well below the threshold we set in $K–S$ test ($p<0.01$). We will say "significantly higher ($p<0.01$)" in the revised version of the paper.

**Table R1: The results of $K–S$ test between the GDIS drought period and the whole period.**

|  | DAP | SDI |
| --- | --- | --- |
| First (0–7 cm) | 2.5e-40 | 8.4e-33 |
| Second (7–28 cm) | 1.0e-21 | 9.3e-22 |
| Third (28–100 cm) | 3.2e-10 | 4.6e-15 |
| Root zone (0–100 cm) | 1.6e-17 | 2.7e-18 |

Additional References

Bodner, G., Nakhforoosh, A. and Kaul, HP.: Management of crop water under drought: a review, Agron. Sustain. Dev., 35, 401–442, https://doi.org/10.1007/s13593-015-0283-4, 2015.

Delbiso, T. D., Altare, C., Rodriguez-Llanes, J. M., Doocy, S., and Guha-Sapir, D.: Drought and child mortality: a meta-analysis of small-scale surveys from Ethiopia, Sci. Rep., 7, https://doi.org/10.1038/s41598-017-02271-5, 2017.

Gasparrini, A., Guo, Y., Hashizume, M., Lavigne, E., Zanobetti, A., Schwartz, J., Tobias, A., Tong, S., Rocklov, J., Forsberg, B., Leone, M., De Sario, M., Bell, M. L., Guo, Y.-L. L., Wu, C., Kan, H., Yi, S.-M., Zanotti Stagliorio Coelho, M. de S., Nascimento Saldiva, P. H., Honda, Y., Kim, H., and Armstrong, B.: Mortality risk attributable to high and low ambient temperature: a multicountry observational study, The Lancet, 386, 369–375, https://doi.org/10.1016/S0140-6736(14)62114-0, 2015.

Hayes, M. J., Wilhelmi, O. V., and Knutson, C. L.: Reducing drought risk: bridging theory and practice, Nat. Hazards Rev., 5(2), 106–113, 2004.

Meyer, E., Aspinwall, M. J., Lowry, D. B., Palacio-Mejía, J. D., Logan, T. L., Fay, P. A., and Juenger, T. E.: Integrating transcriptional, metabolomic, and physiological responses to drought stress and recovery in switchgrass (Panicum virgatum L.), BMC genomics, 15(1), 1–15, 2014.

Shahbazbegian, M., and Bagheri, A.: Rethinking assessment of drought impacts: a systemic approach towards sustainability, Sustain. Sci., 5, 223–236, https://doi.org/10.1007/s11625-010-0110-4, 2010.

---

## Author Response (AR1)

**Reply to Editor**

Dear Rohini Kumar,

Many thanks for your recommendation of revisions and handling our manuscript. We reply to each of the reviewer's comments below. The line number are referring to the revised manuscript with tracked changes. We also provide the revised manuscript and supplementary material (added section "S1 Sensitivity of drought cluster centroids map to the size of drought clusters" as a response to Referee #1).

Sincerely,
Yuya Kageyama and Yohei Sawada

**Reply to Referee #1**

Dear Jakob Zscheischler,

Please find the responses to the comments.

Comments made by the reviewer were highly insightful. They allowed us to greatly improve the quality of the manuscript. We described the response to the comments.

Each comment made by the reviewers is written in italic font. We numbered each comment as (n.m) in which n is the reviewer number and m is the comment number. The line number are referring to the revised manuscript with tracked changes.

Sincerely,

Yuya Kageyama and Yohei Sawada

**Responses to the comments of Referee #1**

*General comments*

*This a valuable contribution as it links drought entries in a recently constructed disaster database (GDIS), which is based on EM-DAT, with actual droughts identified based on a state-of-the-art reanalysis dataset. The work is important because typically disaster databases are not based on information derived from meteorological variables and establishing such links gives credence to the databases but also serves to identify hazard thresholds and differences in vulnerability.*

*Generally the manuscript is very clear and easy to follow but it would benefit from a more in-depth discussion of a number of points that are only briefly mentioned. More detailed comments and suggestions follow below.*
→ Many thanks for the comments and suggestions.

(1.1) *L11: "We found that ERA5-Land soil moisture accurately captured the socio-economic impacts of drought shown in GDIS." I think I understand what the authors mean but as it is this statement is not correct. ERA5-Land cannot capture "impacts of droughts", rather it provides information that droughts actually occurred when disasters labelled as drought-driven where recorded. Similar statements can be found in other places throughout the text. Please adjust the language to more correctly represent your findings and the relationship between drivers of disasters and the recorded impacts.*
→ We have modified the sentence as follows:

    Lines 10–11: "We found that the socio-economic drought impacts shown in GDIS were generally represented by drought hazards quantified from ERA5-Land soil moisture."

(1.2) *L14: "were robust": better us "were less vulnerable"*
→ We appreciate this comment. We have replaced "were robust" with "were less vulnerable."

(1.3) *L112: The authors specify some criteria for the analysed events but it seems the number of analysed events is not affected by this. In line 100 it says "The 282 drought events…were analysed", which suggests that this is the number of events contained in GDIS. In Line 162 it is stated that there are 282 drought events. Please clarify how many events are available in GDIS and how many are finally analysed in this study.*
→ Originally, there are 433 drought events in GDIS and 282 drought events were analyzed. Therefore, the number of analyzed events was indeed affected by our quality control. This point was indeed

unclear in the original version of the paper. We have added this information as follows:

Lines 111–112: "Originally, there are 433 drought events in GDIS and 282 events that met the following criteria were used in this study~."

(1.4) *L176: "We recognized that the drought indices successfully capture the GDIS drought events if the two distributions are not statistically the same." This is only true for the full distribution, not at the individual event level. There could still be many events where no drought is evident from a soil moisture perspective (there is also a strong overlap in the distributions). Furthermore, the test doesn't tell you how the distributions differ. Theoretically it would be possible that all disaster events show less extreme drought conditions compared to the control and the KS test give significant results. Please adjust this statement accordingly and make it more nuanced. This should also be reflected in the abstract.*

→ We fully agree with this comment. In the results section 4.1 (L 226, L 230), we stated that the drought index during the GDIS drought periods was significantly higher than that of the whole period. We checked if the distribution is high/low, as well as the results of $K$–$S$ test.

We have modified the sentence as follows:

Lines 173–175: "We recognized that the GDIS drought events were generally represented by the drought indices quantified from ERA5-Land if the median of the drought index during the GDIS drought periods is higher than that of the whole period and the two distributions of the drought index are not statistically the same."

We have added the sentences as follows:

Lines 233–235: "Note that although we confirmed a general linkage between drought hazards and the GDIS drought events, some GDIS events could not explained by our indices based on the anomaly of soil moisture."

(1.5) *L178: unclear what "socio-economic drought events in GDIS" means. I assume "socio-economic droughts" are the droughts events and impacts recodred in GDIS? Better use something like "drought disasters" or similar*

→ As the reviewer mentioned, "socio-economic droughts" are the drought events and impacts recorded in GDIS. We have replaced "socio-economic drought events in GDIS" with "drought events shown in GDIS", to be consistent with the expressions in other places of our manuscript.

(1.6) *L186: Initially I wasn't completely sure what the purpose of the drought clustering is. I assume the idea is to check whether spatially large droughts typically also lead to impacts (as recorded by EMDAT/GDIS). Given that the drought definition is percentile-based, every location experiences drought with the same frequency, and differences from a rather homogeneous distribution (as for*

*instance visible in the US, Canada and Russia in Fig. 8) are driven by the distribution of continents and the choice of the spatial cutoff (100000km2). With this background, this could be a useful analysis but it would be good to motivate it better and discuss the results in more detail (e.g. it seems that some regions experience more contiguous/large-scale droughts than others, why?). The finding that drought disasters tend to occur in regions that are characterise by frequent large-scale droughts is then quite interesting and novel, especially because from a meteorological perspective and at a pixel level, the frequency of drought occurrence is the same everywhere (20% of the time in this study). So it seems that drought disasters occur when droughts occur over large areas. These findings could be described and discussed in more detail.*

→ As the reviewer mentioned, the idea of drought clustering is to check whether spatially large hydro-meteorological droughts typically lead to disasters that can be seen in GDIS. In the revised version of the paper, we have clarified the role of drought clustering more by the following sentences.

> Lines 327–331: "The consistency between hydro-meteorological drought-prone areas in ERA5-Land and socio-economic drought-prone areas in GDIS shows that spatially large hydro-meteorological droughts (we analyzed at least 100,000 $km^2$) typically lead to impacts as shown in GDIS. Although the drought frequency defined by simulated soil moisture is the same everywhere at the grid level (we set the 20th percentile as a drought threshold), there was considerable heterogeneity in the spatially large drought-prone areas (Fig. 8)."

To reinforce our idea, a sensitivity analysis with different thresholds of the size of drought clusters has been described in the supplement material (see the results in the lowermost part of this response). We have already found that drought-prone areas found in GDIS cannot be reproduced by ERA5-based drought-prone areas when we used too small or large thresholds. We have mentioned it as follows:

> Lines 267–269: "See also the supplement material for sensitivity analysis with different thresholds of the size of drought clusters (Fig. S1), showing that drought-prone areas found in GDIS cannot be reproduced by ERA5-based drought-prone areas when we used too small or large thresholds of the size of drought clusters."

We have also discussed the results more deeply focusing on the underlying climatological mechanisms which are useful to interpret the drought-prone areas. In the revised version of the paper, we have added as follows:

> Lines 331–337: "There are some factors that contribute to the emergence of drought-prone areas, such as El Niño-Southern Oscillation (ENSO), La Niña, intertropical convergence zone (ITCZ), monsoon, land-atmosphere coupling, and anticyclones (Christian et al., 2021). La Niña affects the Horn of Africa, northern China, and western India and has caused severe drought impacts (Funk, 2011; Jain et al., 2021). Ummenhofer et al. (2011) clarified the effect of El Niño–Indian

monsoon relationship on drought in western India. Spatio-temporally large events such as La Niña might cause drought to persist, which leads to drought impacts as shown in GDIS. However, the drought factors are complex, and much future work is needed to reveal the mechanism of the emergence of drought-prone areas. "

**Supplement material**

**S1 Sensitivity of drought cluster centroids map to the size of drought clusters**

Upscaled map of the drought cluster centroids was generated using four thresholds of the size of drought clusters: 10,000 km², 50,000 km², 100,000 km², and 500,000 km², shown in Fig. S1. The drought clusters were generated from the third layer's soil moisture. In the results part, we showed the case of (c) 100,000 km². Drought-prone areas identified by drought indices are not well consistent with those found in GDIS (Fig. 6) if the size of drought clusters is too large or too small. In the case of (a) 10,000 km², there are many more drought-prone areas than those found in GDIS such as western China and Thailand. Although (b) 50,000 km² case shows a similar trend of drought-prone areas with (c) 100,000 km², the drought-prone area in northern China is relatively small. In the case of (d) 500,000 km², drought-prone areas in northern China and western India which are found in (c) 100,000 km² are less outstanding.

[Figure]

**Figure S1: Sensitivity of drought cluster centroids map to the size of drought clusters. (a) 10,000 km², (b) 50,000 km², (c) 100,000 km², and (d) 500,000 km².**

*(1.7) Another point to discuss is that you're using a relative percentile, which means that in generally wet regions, a drought de fined in this way might not be that impactful because the absolute amount*

*of available water is still quite high. This will to some extent also determine where drought disasters occur and might confound the vulnerability assessment. It makes sense to use a relative percentile given that ecosystems/societies are usually adapted to the water availability in their region. However, it is worth discussing this choice.*

→ We appreciate this comment. As an alternative choice, it may be able to use an absolute value of soil moisture as a drought indicator. However, the major limitation of using an absolute value of soil moisture is that it is difficult to conduct a unified drought analysis across multiple regions. As the reviewer mentioned, the thresholds of drought impact occurrences are different in different regions because ecosystems/societies have adapted to the water availability in their region. It is difficult to objectively determine the thresholds in each region. The other limitation is the biases of absolute values of soil moisture in reanalysis products. Many climate studies have used relative values rather than absolute values because biases in climate models are less important in relative values (Liu and Key, 2016).

The limitation of a relative percentile-based drought quantification is that the drought in extremely wet regions might not be accurately represented as the reviewer points out. In the drought clustering analysis, we showed that Indonesia was hydro-meteorological drought-prone areas in ERA5-Land, which were not included in drought-prone areas found in GDIS. We attributed this inconsistency to the abundance of absolute water availability in Indonesia.

The issues discussed above have been included as follows:

Lines 360–371: "We used the percentile soil moisture, deviation from the normal condition, to quantify drought following many previous studies (e.g., Sheffield and Wood, 2011; Hanel et al., 2018). However, the inconsistency for Indonesia between hydro-meteorological drought-prone areas and drought-prone areas found in GDIS implies that the drought in extremely wet regions might not be well represented. It implies that our drought quantification method based on relative values of soil moisture cannot accurately consider the amount of regularly available water resources. An alternative way to quantify drought is to use an absolute soil moisture value, but it is not straightforward to quantify drought events by absolute soil moisture values. The thresholds of drought impact occurrences in absolute values are different in different regions, because ecosystems/societies have adapted to the water availability in their region. It means that a unified drought analysis across multiple regions is difficult to be developed based on absolute values of soil moisture. The other limitation is the biases of absolute values of soil moisture in reanalysis products. Many climate studies have used relative values rather than absolute values because biases in climate models are less important in relative values (Liu and Key, 2016)."

(1.8) *L224: You could mention some more details in the results. For instance, how large are the differences between the medians for the different soil moisture layers in Figs 3 and 4. I assume the SDI for the whole period is approx 1 by definition (Fig. 4)? I would also mention that for clarification.*
→ We have described how large are the differences. The mean value of SDI for the whole period is approximately 1 by the definition (Fig. 4).

(1.9) *L239: It maybe worth checking how these regions differ in their absolute SM values (averaged over time). Regions with lower water availability in general might experience drought disasters more frequently even though the relative deviation from normal conditions is small (see the comment higher up). Independent of the findings, the interpretation that the identified regions are more vulnerable to drought probably still holds, but I think it can slightly change the interpretation and consequences for resilience planning. If water is typically abundant, it's much easer to be drought resistant.*
→ We appreciate this comment. We have replaced Fig. 5 with the one shown below, whose colors of the violins show the average absolute soil moisture value in each region. Sub-Saharan Africa, which was vulnerable to drought, shows lower water availability. It may be one of the reasons for the difficulty in managing the drought hazards in Sub-Saharan Africa. We have added this point as follows:

> Lines 296–298: "As shown in Fig. 5, Sub-Saharan Africa, which was vulnerable to drought, showed lower water availability. It may be another reason for the difficulty in managing the drought hazards in Sub-Saharan Africa."

[Figure]

**Figure 5: Comparison of the root-zone SDI by geographical regions. The red line shows the median value and grey dotted lines show the 25th and 75th percentile values of each distribution. The colour shows the average soil moisture over the study period (1964–2018).**

*(1.10) L282: Note that the results of Fig. 5 also confirm results from Tschumi & Zscheischler (2020) who also found smaller climate anomalies in less developed countries during EMDAT disasters for different climate variables (their Fig. 9).*

→ We appreciate this comment. We have discussed that our result is consistent with this work in the revised version of the paper as follows:

> Lines 285–287: "Tschumi and Zscheischler (2020) also showed smaller climate anomalies in less developed countries associated with EM-DAT disasters, meaning that less developed countries were vulnerable to natural hazards, as shown in our Fig. 5."

*(1.11) L310: unclear what "reproducibility of ERA5-Land" means. Please clarify.*

→ "Reproducibility of ERA5-Land" means the performance of ERA5-Land to simulate soil moisture. We have modified the sentence as follows:

> Line 310: "The performance of ERA5-Land to simulate soil moisture might affect these inconsistencies."

*(1.12) L340: again, unclear what "reproducibility" of reanalysis products means. It seams the authors mean that using different climate datasets, one obtains similar drought maps (which makes sense) whereas this does not hold for socioeconomic impacts. This of course depends on how these things are defined. If the same drought definition is used, the choice of (climate) dataset has only little effect on the analysis as this information is comparably well constrained by observations. For socio-economic impacts, no clear/objective definition of disasters etc. exists and uncertainties in impact estimates are usually high. So a comparison across datasets is difficult also because different datasets use different definitions.*

→The "reproducibility" (L340 in the revised manuscript with tracked changes) means the performance of reanalysis products to simulate hydrometeorological variables. We believe that the validation of the performance has not been fully conducted in terms of whether the anomalies of simulated variables are consistent with the socio-economic impact of drought events in disaster databases. This "reproducibility" does not imply the similarity of results with different reanalysis products. We believe that this comment is provided because we failed to deliver this point using the misleading word of "reproducibility". In the revised version of the paper, this wording is no longer used. We have modified the sentences as follows:

> Lines 339–345: "Although various reanalysis products have been developed and their validations have been conducted by comparing them with earth observation data (e.g., Muñoz-Sabater et al., 2021; Reichle et al., 2017; Rodell et al., 2004), few studies have examined the validation in terms of the disaster occurrence. Sawada (2018) compared the areas identified as drought from a reanalysis product with the disaster records from EM-DAT, but only in a country-scale. As seen

in Fig. 7, national-level information does not provide accurate views of disaster locations, so that it is insufficient for validation data. The use of sub-national disaster databases such as GDIS opens the door to validate reanalysis products in terms of the disaster occurrence."

(1.13) *L403: "GDIS only covers about 60% of droughts in EM-DAT" Can you comment on why this is the case? Was more detailed information on the remaining events not available?*
→ This is due to vague or unknown location names in EM-DAT. We have added this point as follows: Lines 403–404: "GDIS only covers about 60% of droughts in EM-DAT due to vague or unknown location names in EM-DAT."

Additional References

Christian, J., I., Basara, J. B., Hunt, E. D., Otkin, J. A., Furtado, J. C., Mishra, V., Xiao, X., and Randall, R. M.: Global distribution, trends, and drivers of flash drought occurrence, Nat. Commun., 12, https://doi.org/10.1038/s41467-021-26692-z, 2021.

Funk, C.: We thought trouble was coming, Nature, 476, 7, https://doi.org/10.1038/4760007a, 2011.

Jain, S., Mishra, S. K., Anand, A., Salunke, P., and Fasullo, J. T.: Historical and projected low-frequency variability in the Somali Jet and Indian Summer Monsoon, Clim. Dynam., 56, 749–765, https://doi.org/10.1007/s00382-020-05492-z, 2021.

Liu, Y. and Key, J. R.: Assessment of Arctic Cloud Cover Anomalies in Atmospheric Reanalysis Products Using Satellite Data, J. Climate, 29, 6065–6083, https://doi.org/10.1175/JCLI-D-15-0861.1, 2016.

Palecki, M. A.: Using Standardized Soil Moisture Indices for Drought Monitoring, in: AGU Fall Meeting Abstracts, H53J-1725, 2018.

Ummenhofer, C. C., Sen Gupta, A., Li, Y., Taschetto, A. S., and England, M. H.: Multi-decadal modulation of the El Nino-Indian monsoon relationship by Indian Ocean variability, Environ. Res. Lett., 6, https://doi.org/10.1088/1748-9326/6/3/034006, 2011.

**Reply to Referee #2**

Dear Mariana Madruga de Brito,

Please find the responses to the comments.

Comments made by the reviewer were highly insightful. They allowed us to greatly improve the quality of the manuscript. We described the response to the comments.

Each comment made by the reviewers is written in italic font. We numbered each comment as (n.m) in which n is the reviewer number and m is the comment number. The line numbers are referring to the revised manuscript with tracked changes.

Sincerely,
Yuya Kageyama and Yohei Sawada

**Responses to the comments of Referee #2**

*In this manuscript, the authors use the new GDIS data and link it to hazards (i.e. soil moisture). The contribution is timely and innovative. I enjoyed reading the paper. Currently, we have little understanding of the relationships between impacts and hazard data. Hence, the proposed approach provides a way to understand these linkages. I particularly appreciate that instead of the traditional approach (e.g. hazard x vulnerability x exposure = risk), the authors go from the disaster itself, and from it try to identify the hazard drivers.*

*The manuscript is well written, yet, some issues need to be addressed:*
→ Many thanks for the comments and suggestions.

(2.1) *My main criticism is that the authors claim to have assessed the "regional vulnerability to drought" (see line 283). Figure 5 shows that the SDI alone cannot explain drought disasters. Indeed, in Sub-Saharan Africa and South Asia, the SDI is not as high as in Europe and North America, yet, many disasters are observed. One explanation for this is indeed, the socio-economic vulnerability in these regions is higher. However, it could also be that it is the population exposure. Since you have not accessed any of these, you cannot make claims regarding the vulnerability.*
→ We appreciate this comment. As the reviewer mentioned, we did not directly include exposure, and "regional vulnerability to drought" was overstated. There are several difficulties in treating exposure in our study. To estimate exposure, it is necessary to identify what has been damaged (e.g., people, crops, forests, etc.). However, EM-DAT provides no information on impact types (e.g., water shortage, famine, wildfire, etc.) in many drought events, which inhibits the identification of what has been damaged. In addition, EM-DAT provides no information about the amount of damage in multiple drought events (line 406). Therefore, we have changed "regional vulnerability to drought" to "proxy of regional vulnerability to drought". In addition, we have explicitly said that our analysis did not directly include exposure in the discussion (See also our response to the comments 2.4 and 2.7).

(2.2) *You mentioned that you disregarded drought events that lasted for less than 2 months. However, I assume most of the EM-DAT data only has a year, and few have a start and end date. In such cases, you mentioned that you considered the whole year as a drought year. That can be very problematic, as probably many of these events could actually be 2 months. It would be great if you could bring this limitation in the discussion section, where you already mentioned the limitation of the impact data.*
→ We fully agree with this comment. There are some GDIS drought events in which the details of drought duration information are not provided. Therefore, drought events shorter than two months may be included in our analysis, although we intended to exclude them. We have added this limitation

as follows:

> Lines 408–410: "Although we excluded GDIS drought events shorter than two months from our analysis, some of the analyzed events might be shorter than two months. This is because we applied January for the start and December for the end of the event if the start and/or end months of events shown in EM-DAT were unclear."

(2.3) *Why was landcover data from 2020 used (line 127) if the drought events go from 1964 to 2018? Likewise, why did you consider soil moisture data from 2019-2020 (line 132) when your impact data is from previous years?*

→ The detailed landcover datasets such as MODIS landcover (used in our study), GlobCover, and Global Land Cover Characterization (GLCC) do not provide landcover data over a long period of time. The MODIS land cover dataset starts in 2001. In addition, we only used landcover data to exclude the barren or sparsely vegetated areas from drought-prone areas by the drought clustering method, so that the timing of land cover does not significantly change our results. We confirmed that there was little or no difference in the extent of the barren or sparsely vegetated areas when we compared with the oldest 2001 data and the 2020 data. We have decided not to change the paper responding to this point.

We used (absolute) soil moisture data during 1950–2020 in calculating the percentiles. After the percentiles were calculated, only (percentile) data during the period of the GDIS drought events (1964–2018) was used (line 131). We expected that a longer data period would contribute to more robust percentile values. For calculating SPI, at least 30 years of data is required (McKee et al., 1993). We also used percentile values which were calculated from absolute values during 1964–2018 (the same period with the GDIS drought events) and the results were slightly better when used percentile values which were calculated from absolute values during 1950–2020; the differences of drought indices during the GDIS drought period and the whole period was bigger and hydro-meteorological drought-prone areas were more consistent with GDIS drought-prone areas (not shown). We have added a comment as follows:

> Lines 140–142: "We used the longer period of original ERA5-Land data (1950–2020) to calculate percentiles than the study period (1964–2018) to yield more robust percentile values."

(2.4) *Item 3.3 and 4.2. Similar to the first comment, stratifying your SDI data by different regions is not equivalent to "understanding the vulnerability in each region". What you did here was to analyze the SDI according to different regions. This is definitely an important analysis, but it is not a vulnerability analysis. You can hypothesise that the vulnerability causes these differences, but you cannot confirm this with the present data. The statements in line 255 are pretty strong and cannot be made without considering the socio-economic vulnerability. I suggest toning them down and writing*

*that "the vulnerability could explain these differences". However, for this, further analysis is required.*
→ We appreciate this comment. Relating to the first comment (2.1), we have toned down like "vulnerability could explain these differences." We have also changed the subtitle "Regional vulnerability to drought" to "Regional levels of drought indices associated with GDIS drought events". The detailed comment on exposure has been described in the discussion section (See also comment 2.7). Below is the revised version of these sections.

**3.3 Regional levels of drought indices associated with GDIS drought events**
The levels of hydro-meteorological drought indices associated with drought events shown in GDIS are different in different regions. Vulnerability could explain these differences (Delbiso et al. 2017; Gasparrini et al. 2015; Tschumi and Zscheischler, 2020). Note that vulnerability is not the only explanation for these differences; exposure is another factor that influences the linkage between hazards and impact (Visser et al., 2014; see also the discussion section). Since we did not directly include exposure, we recognized these differences as "the proxy of vulnerability". Following Bachmair et al. (2016), and Tschumi and Zscheischler (2020), the levels of SDI which are associated with socio-economic drought events in GDIS were quantified and analyzed. The levels of SDI were stratified by geographical regions to understand the distribution of the proxy of vulnerability in each region.

**4.2 Regional levels of drought indices associated with GDIS drought events**
Figure 5 shows ~. Having many small SDI events indicates that less severe hydro-meteorological droughts have caused serious socio-economic impacts, meaning that the regions are vulnerable to drought. On the other hand, the regions with many large SDI events can be recognized as robust regions to drought. Thus, Sub-Saharan Africa and South Asia are vulnerable to drought, while North America and Europe are less vulnerable to drought. This regional characteristic of the proxy of vulnerability to drought can be found when SDI is generated by soil moisture in different soil layers (not shown). Note that Middle East & North Africa were excluded from the analysis because the sample size was too small (n = 4).

In the introduction (line 78), we have modified "we quantified the vulnerability to drought in different geographical regions" as follows:

> Lines 76–78: "Then, we quantified the levels of drought indices associated with GDIS drought events in different geographical regions, which could benefit the quantification of vulnerability to drought".

*(2.5) Item 3.4. It is not clear to me what the advantage of clustering is. Why would the results be different with or without clustering? Could you please elaborate?*

→ The idea of drought clustering is to check whether spatially large hydro-meteorological droughts typically lead to disasters that can be seen in GDIS. For more detail, please see our responses to the comment of the reviewer #1 (comment 1.6).

*(2.6) In line 287–289, you say that your findings are consistent with previous ones that used GDP, however, you have not used GDP here. Please rephrase the sentence, or do the analysis*

→ We intended to say that high-income countries were associated with lower vulnerability to drought. We did not use GDP, as the reviewer mentioned. We have modified the sentences as follows:

> Lines 287–289: "Previous studies have shown that higher GDP per capita is associated with lower vulnerability to natural hazards (e.g., Kim et al., 2019; Tanoue et al., 2016). North America and Europe are high-income countries, and these previous works support our findings."

*(2.7) In line 294−295, you write that the vulnerability is explained by the lack of structural measures. Sure, these are contributing factors, but socio-economic variables and exposure could be even more important. I understand the author's point of view, being an engineer myself. However, "command and control" approaches in which drought risk is controlled solely with conventional engineering measures is an outdated view. The authors cannot ignore the vast literature on socio-economic vulnerability and risk perception. You do not need to cite these papers, but have a look at the GAR chapters on vulnerability and droughts:*

*https://gar.undrr.org/sites/default/files/chapter/2019-05/Chapter_3.pdf*

*https://gar.undrr.org/chapters/chapter-6-special-section-drought*

→ We appreciate this comment. We stated the complex nature of vulnerability in the discussion sections (from line 319) and included non-structural countermeasures. We have strengthened the content including exposure in this section as follows:

> Lines 390–398: "Exposure is another important factor that influences the linkage between hazards and impact (Visser et al., 2014). The inconsistency between hydro-meteorological drought-prone areas in ERA5-Land and drought-prone areas found in GDIS in Namibia implies that the quantification of exposure is necessary to strengthen the analysis in our study. It is necessary to identify what has been damaged (e.g., people, crops, forests, etc.) to quantify exposure. However, EM-DAT provides no information about impact types (e.g., water shortage, famine, wildfire, etc.) in many drought events, which inhibits the identification of what has been damaged. Like vulnerability, exposure is complex and dynamic. For example, the level of exposure is affected by changes in crop growing with the seasons (Bodner et al., 2015). To

improve our analysis on vulnerability shown in Sect. 4.2, detailed analyses on the complex and dynamic nature of both vulnerability and exposure are necessary."

(2.8) *Line 313: Spain is not shown in Figure 5. Hence, you cannot make this statement based only in the Figure. It could well be that the other European countries are "robust" and push the mean up, whereas Spain pushes the mean down. I suggest removing the sentence or doing the analysis for Spain only.*
→ Spain is actually less vulnerable to drought. We have added the mean value of SDI for Spain in the manuscript as follows:

> Lines 313–314: "Spain, a member of European countries, is less vulnerable to drought, as shown in Fig. 5 (two events were observed in Spain, and their average SDI was 4.3)."

(2.9) *Line 346: I find it good that you can find good linear correlations. However, I was surprised as soil moisture and drought impacts usually do not have a linear relationship. Indeed, there are non-linear relationships (there is a lag between soil moisture and forestry impacts for instance). Furthermore, you have multi-year drought events, which add complexity to the interpretation of spatial and temporal differences in drought prevalence. Indeed, it could be that a drought is not so severe in biophysical terms. However, because they last so long, the impacts are high. I assume you got good correlations because EMDAT usually do not have long term impacts but rather focuses on immediate ones. Please add a discussion of these issues in your discussion section.*
→ We fully agree that the relationship between soil moisture and drought processes is non-linear. As the reviewer mentioned, there are non-linear relationships between the drought severity and drought stress in vegetation (e.g., Chen et al. 2020; Meyer et al., 2014), where damage increases suddenly when the drought severity exceeds a certain critical threshold. Note that we found no "good linear correlations", nor did we explore any linear relationships in this paper. We did not explicitly describe that our analyzed relationship is linear.

We agree that there is a lag between hazards and impact. Whether EM-DAT captures the long-lasting drought impacts is an important issue. The reason why drought events shown in GDIS were generally represented by drought hazards quantified from ERA5-Land may be due to the relatively long period of the drought duration in GDIS; the mean duration is approximately 12 months. Drought phenomenon is complex; the impacts of drought last even after the hydro-meteorological drought ends. Further analysis is needed to focus on the chronological correspondence to drought hazards. We have added the following discussion emphasizing the complex nature of the relationship between soil moisture and drought impacts as follows:

Lines 375–384: "The relationship between hazards and impact is much more complex than addressed in this study. Many studies have revealed the non-linear relationships between the drought severity and the reduction of vegetation growth (e.g., Chen et al. 2020; Meyer et al., 2014), where damage increases suddenly when the drought severity exceeds a certain critical threshold. On the other hand, de Brito et al. (2020) reported that there was a linear relationship between the drought severity and the number of drought articles as a proxy of socio-economic drought impacts. In addition, drought is a long-lasting disaster and there is a time-lag between hazards and impact, so that the period of hydro-meteorological drought is not necessarily consistent with the period considered as a disaster in EM-DAT. Some studies have revealed that the impacts of drought last even after the hydro-meteorological drought ends (e.g., Shahbazbegian and Bagheri, 2010). Yokomatsu et al. (2020) analyzed the impact of the drought in terms of the economic development after the drought. In any case, further analyses are needed to focus on the chronological correspondence to drought hazards."

(2.10) *Line 9: What is meant by "drought information" here? Do you mean Drought impact information (i.e., the GDIS data?). Please be more specific.*
→ This is drought impact information as shown in GDIS. We have changed the language.

(2.11) *Line 58-60: I do not understand the purpose of these 3 last sentences. When you write about vegetation growth, do you mean forest or crops? Regarding the last sentence. Why do we need to treat drought impacts as they are socially perceived? If you think the section is relevant, please expand it to make it clearer to the reader.*
→ Vegetation means plants in general, both forests and crops. We intended to say that not all droughts affect vegetation growth, and not all vegetation decline is caused by drought. "Socially perceived as drought" implies that we should consult a disaster database, which shows events in which the society has actually suffered from drought (we say it as "socially perceived"). We have modified the sentences as follows:

Lines 58–60: "It is unclear whether socio-economic drought impacts are associated with declined vegetation growth. It is ideal to treat the socio-economic drought impact based on a disaster database since it directly shows events in which the society has actually suffered from drought."

(2.12) *Line 100: This information is repeated in the introduction. I suggest removing it: "GDIS is the geocoded...."*
The next sentence is about EM-DAT, which made us say "GDIS is the geocoded disaster locations database based on EM-DAT". We have modified the sentence as follows:

Line 100: "GDIS is generated based on EM-DAT."

(2.13) *Line 128−131: You do not "show the drought vulnerability". Land cover is an exposure data, not vulnerability.*

→ We have changed "show the drought vulnerability" to "show the levels of drought indices associated with GDIS drought events." We intended to use land cover data as exposure data. We have modified the sentences as follows:

> Lines 128–131: "To show the levels of drought indices associated with GDIS drought events by geographical regions, we used the classification of the world bank geographical regions.
> As a proxy of exposure data, we used the MODIS land cover~."

(2.14) *Line 226: "significantly higher": could you please provide the test results that made you reach this conclusion.*

→ The test results are below the table. In any case, the values are well below the threshold we set in $K−S$ test ($p<0.01$). We have modified the sentence as follows:

> Line 226: "significantly higher ($p<0.01$)"

**Table R1: The results of *K–S* test between the GDIS drought period and the whole period.**

|  | DAP | SDI |
| --- | --- | --- |
| First (0–7 cm) | 2.5e-40 | 8.4e-33 |
| Second (7–28 cm) | 1.0e-21 | 9.3e-22 |
| Third (28–100 cm) | 3.2e-10 | 4.6e-15 |
| Root zone (0–100 cm) | 1.6e-17 | 2.7e-18 |

Additional References

Bodner, G., Nakhforoosh, A. and Kaul, HP.: Management of crop water under drought: a review, Agron. Sustain. Dev., 35, 401–442, https://doi.org/10.1007/s13593-015-0283-4, 2015.

Delbiso, T. D., Altare, C., Rodriguez-Llanes, J. M., Doocy, S., and Guha-Sapir, D.: Drought and child mortality: a meta-analysis of small-scale surveys from Ethiopia, Sci. Rep., 7, https://doi.org/10.1038/s41598-017-02271-5, 2017.

Gasparrini, A., Guo, Y., Hashizume, M., Lavigne, E., Zanobetti, A., Schwartz, J., Tobias, A., Tong, S., Rocklov, J., Forsberg, B., Leone, M., De Sario, M., Bell, M. L., Guo, Y.-L. L., Wu, C., Kan, H., Yi, S.-M., Zanotti Stagliorio Coelho, M. de S., Nascimento Saldiva, P. H., Honda, Y., Kim, H., and Armstrong, B.: Mortality risk attributable to high and low ambient temperature: a multicountry observational study, The Lancet, 386, 369–375, https://doi.org/10.1016/S0140-6736(14)62114-0, 2015.

Meyer, E., Aspinwall, M. J., Lowry, D. B., Palacio-Mejía, J. D., Logan, T. L., Fay, P. A., and Juenger, T. E.: Integrating transcriptional, metabolomic, and physiological responses to drought stress and recovery in switchgrass (Panicum virgatum L.), BMC genomics, 15(1), 1–15, 2014.

Shahbazbegian, M., and Bagheri, A.: Rethinking assessment of drought impacts: a systemic approach towards sustainability, Sustain. Sci., 5, 223–236, https://doi.org/10.1007/s11625-010-0110-4, 2010.